# Linear Speedup in Personalized Collaborative Learning

## Abstract

Collaborative training can improve the accuracy of a model for a user by trading off the model's bias (introduced by using data from other users who are potentially different) against its variance (due to the limited amount of data on any single user). In this work, we formalize the personalized collaborative learning problem as a stochastic optimization of a task 0 while given access to $N$ related but different tasks $1, \ldots, N$. We give convergence guarantees for two algorithms in this setting—a popular collaboration method known as *weighted gradient averaging*, and a novel *bias correction* method—and explore conditions under which we can achieve linear speedup w.r.t. the number of auxiliary tasks $N$. Further, we also empirically study their performance confirming our theoretical insights.

## 1 Introduction

Collaborative learning is the setup where agents/users/clients collaborate in hope of better performance (faster convergence, smaller inference time, or generalization) compared to each agent working alone. Federated learning and auxiliary learning are two examples of collaborative learning. In Federated Learning multiple users train a machine learning model on their combined datasets (Kairouz et al., 2019). Collaboration vastly increases the amount of data available for training. However, the other users may be heterogeneous, i.e., they may have datasets and objectives which do not match those of the considered user. Combining data from such heterogeneous users can significantly hamper performance, with even worse performance than when training alone (Yu et al., 2020).

Training alone and on combined data represent two extremes, with the former having no bias but high variance and the latter having low variance but high bias. Alternatively, personalized collaborative learning algorithms (where each user only cares about its own performance) (Wang et al., 2019; Mansour et al., 2020) attempt to find 'in-between' models that trade off some bias against variance. In the best case, we can use the data from the $N$ other users to reduce our variance by a factor $N$ (called *linear speedup*) while simultaneously not incurring any bias.

In auxiliary Learning, the goal is to train one main task by combining it with auxiliary tasks that are related in some sense to the main task. In this sense, auxiliary learning can be seen as more general than personalization in Federated Learning.

In this work, we explore from a purely theoretical lens, under what conditions a given agent can benefit from personalized collaborative learning. We consider an idealized scenario where the goal is optimizing a fixed user's stochastic function $f_0(\boldsymbol{x})$, while also giving access to stochastic gradients of $N$ other collaborators $\{f_1(\boldsymbol{x}), \ldots, f_N(\boldsymbol{x})\}$. We also neglect communication issues: the users can be all on the same server for example or the collaborators can be treated as auxiliary functions. In the latter case, one important question is to know how much can we benefit from such auxiliary "information" available to us (maybe for free). We start with the simple strategy of **weighted gradient averaging** that uses a weighted average of the gradient estimates as a pseudo-gradient and then takes an SGD step. We show that while there do exist scenarios where this simple strategy suffices, it can also incur significant bias introduced by the collaborators. This then motivates our main method of **bias correction** which uses the past observed gradients to estimate and correct for these biases. We show that our proposed solution solves the problems WGA had with bias. Furthermore, we get a linear speedup in the number of agents that satisfy a mild dissimilarity constraint.

**Contributions.** Our main contributions include:

- Formalizing the collaborative stochastic optimization problem where an agent is required to minimize their objective by collaborating with other agents, in contrast to traditional federated learning.
- Proving convergence rates for *weighted gradient averaging* and proposing and analyzing a novel *bias correction* algorithm.
- Showing that with the correct choice of hyper-parameters and under a mild condition on the dissimilarity between agents, bias correction enjoys a linear speedup in the number of (relatively similar) collaborators, with variance reducing as collaborators increase (and a bias going to zero in the number of steps).

## 2  Related Work

**Federated and Decentralized Learning.** Federated learning (FL) (Konecny et al., 2016; McMahan et al., 2017; Mohri et al., 2019) denotes a machine learning setting where a global set of training data is distributed over multiple users (also called agents or clients). These users form a 'federation' to train a global model on the union of all users' data. The training is coordinated by a central server, and the users' local data never leaves its device of origin. Owing to data locality and privacy awareness, FL has become prominent for privacy-preserving machine learning (Kairouz et al., 2019; Li et al., 2020a; Wang et al., 2021). Our studied setting is different because we learn the objective of one specific user, not the union of users. *Decentralized learning* refers to the analogous more general setting without a central server, where users communicate peer-to-peer during training, see e.g. (Nedic, 2020).

**Personalization.** Due to device heterogeneity, and data heterogeneity, a 'one model fits all approach leads to poor accuracy on individual users. Instead, we need to learn personalized models for each user. Prominent approaches for this include performing additional local adaptation or fine-tuning (Wang et al., 2019; Fallah et al., 2020), or weighted averaging between a global model and a locally trained model (Mansour et al., 2020; Deng et al., 2020; Hanzely & Richtárik, 2020). Collins et al. (2020); Khodak et al. (2019) investigate how such local fine-tuning can improve performance in some simple settings if the users' optima are close to each other. In another highly relevant line of work, Maurer et al. (2016); Tripuraneni et al. (2020); Koshy Thekumparampil et al. (2021); Feng et al. (2021) shows how a shared representation can be leveraged to perform efficient transfer of knowledge between different tasks (and users). Li et al. (2020c); Mohri et al. (2019); Yu et al. (2020) investigate how FL distributes the accuracy across users and show that personalization gives a more equitable distribution. We refer to (Kulkarni et al., 2020) for a broader survey of personalization methods.

Unlike most of the above works, we consider the perspective of a single agent/user . Further, while our weighted gradient averaging is closely related to weighted model averaging, the bias correction method is novel and is directly motivated by our theory. Finally, while several of the above works (e.g. Mansour et al., 2020; Deng et al., 2020) also provide theoretical guarantees, they use a statistical learning theory viewpoint whereas we use a stochastic optimization lens.

Perhaps the works closest to ours are (Donahue & Kleinberg, 2020) and (Grimberg et al., 2021), both of whom study model averaging. The former uses game theory to investigate whether self-interested players have an incentive to join an FL task. This is true as long as users achieve significantly better performance when training together than when training alone. Their work further highlights the importance of understanding when personalization can improve performance. More recently, Grimberg et al. (2021) consider a weighted model averaging of two users for mean estimation in $1D$. Both these works study only toy settings with restrictive assumptions. Our results are more general and include non-convex optimization.

More recently there was an attempt to formalize a new selfish variant of Federated Learning (Ruichen et al., 2022), a new setting where we only care about the performance of a subset of internal clients all the while using/collaborating with external clients. This setting is a particular case of the one considered here (by taking client 0 to be the average of internal clients). Also, (Mestoukirdi et al., 2021) proposes a user-centric formulation of federated learning that can be seen as a particular case of our weighted gradient averaging scheme, further they show empirically that communication load problems can be overcome by clustering agents. The last two works lack rigorous theory to back their results.

**Auxiliary learning.** More generally, there is a framework that combines a main task with auxiliary tasks in order to minimize the performance of the main task; this is usually done by minimizing a weighted linear combination of the main loss with the auxiliary losses which is similar to WGA. In most of the works in this area, there is a lack of convergence guarantees. The auxiliary tasks or collaborators as we call them can

be seen as (hopefully more informative) priors as in (Baifeng et al.). Such works that use this idea are for example (Xingyu et al.) for reinforcement learning (which optimizes the collaboration weights as well) and (Aviv et al.) which considers a general combination (not necessarily linear) implemented by a neural network and performs optimization via implicit differentiation. All these works are based on approximations and lack convergence guarantees as stated before. In this work, we propose a simpler model (constant collaboration weights) but analyze rigorously the convergence of all algorithms which has not been done before.

Lately, Chayti & Karimireddy (2022) took inspiration from stochastic variance reduction methods (Johnson & Zhang, 2013) to propose a way to perform optimization when having access to auxiliary information and give theoretical guarantees, however, this work does not have the linear speedup in terms of the number of collaborators that we have in this work.

**Control variates.** There is some similarity between our bias correction method and other control variate methods such as SCAFFOLD (Karimireddy et al., 2019), however the local vs global objectives as well as the resulting updates are different. Also, we use an exponential moving average whereas other control variates use mainly an SVRG-like correction (for a detailed discussion see Appendix A.2).

## 3 Setup and Assumptions

In this section, we formalize personalized collaborative optimization and discuss our assumptions.

### 3.1 Personalized Collaborative Stochastic Optimization

We model collaborative optimization as an environment where $N + 1$ users denoted $0, \ldots, N$ can interact with each other. Each user $k$ has only access to its own objective $f_k(\boldsymbol{x}) := \mathbb{E}_{\xi^{(k)}}[f_k(\boldsymbol{x}; \xi^{(k)})]$ (e.g. a loss function evaluated on their own data), where $\xi^{(k)}$ is a random variable from which we can sample without necessarily knowing its distribution (this covers the online optimization setting as well as optimizing over finite training data sets). The users can collaborate by sharing (stochastic) gradients that they compute on their private loss function $f_k$ on a shared input parameter $\boldsymbol{x}$.

We formalize the personalized collaborative stochastic optimization problem as solving for user 0's goal:

$$\min_{\boldsymbol{x} \in \mathbb{R}^d} f_0(\boldsymbol{x}) \,, \tag{1}$$

by exchanging gradients with the other users. This exchange of information between the main user '0' and their collaborators can be done in many ways. In this work, to solve problem (1), user 0 updates their state $\boldsymbol{x}_t$ by using different variants of a gradient estimate $\boldsymbol{g}(\boldsymbol{x}_t)$ and step size $\eta_t$:

$$\boldsymbol{x}_{t+1} = \boldsymbol{x}_t - \eta_t \boldsymbol{g}(\boldsymbol{x}_t) \,. \tag{2}$$

As illustrated in Algorithm 1, each collaborator $k$ computes an unbiased local gradient estimate $\boldsymbol{g}_k(\boldsymbol{x}_t) := \nabla_{\boldsymbol{x}} f_k(\boldsymbol{x}_t; \xi_t^{(k)})$ of $\nabla_{\boldsymbol{x}} f_k(\boldsymbol{x}_t)$ at $\boldsymbol{x}_t$, and shares those with the main user 0. Using these helper gradients as well as its own gradient, user 0 then forms the final $\boldsymbol{g}(\boldsymbol{x}_t)$ and takes an update step.

The simplest baseline to consider (henceforth called the 'Alone' method) is the case where user 0 ignores the collaborators and decides to work alone by setting $\boldsymbol{g}(\boldsymbol{x}_t) = \boldsymbol{g}_0(\boldsymbol{x}_t)$. In general, $\boldsymbol{g}(\boldsymbol{x}_t)$ can be formed in several different ways, using current gradients as well as past gradients.

### 3.2 Assumptions

**Notation.** For each user $k$, we denote by $x_k^\star$ a stationary point of $f_k$, and $f_k^\star$ its corresponding value. We denote the gradient noise $\boldsymbol{n}_k(\boldsymbol{x}, \xi) = \boldsymbol{g}_k(\boldsymbol{x}_t) - \nabla_{\boldsymbol{x}} f_k(\boldsymbol{x}_t)$.

We make the following common assumptions:

*A1 (Smoothness)* $f_0$ is $L$ smooth, i.e., $\forall \, \boldsymbol{x}, \boldsymbol{y} \in \mathbb{R}^d$:

$$\|\nabla_{\boldsymbol{x}} f_0(\boldsymbol{x}) - \nabla_{\boldsymbol{x}} f_0(\boldsymbol{y})\| \leq L\|\boldsymbol{y} - \boldsymbol{x}\| \,.$$

---

**Algorithm 1** Collaborative Stochastic Optimization

---
**Require:** Collaborators $k = 0, \ldots, N$
**Require:** $\boldsymbol{x}_0$; $\eta_t$; $T$
    **for** $t = 0 \ldots T - 1$ **do**
        **for all** users $k = 0, \ldots, N$ **in parallel do**
            Sample $\xi_t^{(k)}$
            Compute $\boldsymbol{g}_k(\boldsymbol{x}_t) := \nabla_{\boldsymbol{x}} f_k(\boldsymbol{x}_t; \xi_t^{(k)})$
        **end parfor**
    $\nabla$ Aggregation on user 0:
        Form $\boldsymbol{g}(\boldsymbol{x}_t)$ using received $\{\boldsymbol{g}_k(\boldsymbol{x}_{t'})\}_{k=0,\ldots,N}^{t' \leq t}$
        $\boldsymbol{x}_{t+1} = \boldsymbol{x}_t - \eta_t \boldsymbol{g}(\boldsymbol{x}_t)$
    **end for**     **return** $\boldsymbol{x}_T$

---

*A2 ($\mu$-PL)* $f_0$ satisfies the $\mu-$PL condition, i.e.:

$$\forall\, \boldsymbol{x} \in \mathbb{R}^d : \|\nabla_{\boldsymbol{x}} f_0(\boldsymbol{x})\|^2 \geq 2\mu(f_0(\boldsymbol{x}) - f_0^\star).$$

And for each agent $k \in \{0, \ldots, N\}$:
*A3 ($\delta$-Bounded Hessian Dissimilarity, or $\delta$-BHD)*

$$\forall\, \boldsymbol{x} \in \mathbb{R}^d : \|\nabla_{\boldsymbol{x}}^2 f_k(\boldsymbol{x}) - \nabla_{\boldsymbol{x}}^2 f_0(\boldsymbol{x})\| \leq \delta.$$

*A4 (Gradient Similarity)* $\exists\, m, \zeta_k^2 \geq 0$ s.t. $\forall \boldsymbol{x} \in \mathbb{R}^d$:

$$\|\nabla_{\boldsymbol{x}} f_k(\boldsymbol{x}) - \nabla_{\boldsymbol{x}} f_0(\boldsymbol{x})\|^2 \leq m\|\nabla_{\boldsymbol{x}} f_0(\boldsymbol{x})\|^2 + \zeta_k^2.$$

*A5 (Bounded variance)* $\exists\, \sigma_k^2 \geq 0$ s.t. $\forall \boldsymbol{x} \in \mathbb{R}^d$:

$$\mathbb{E}[\|\boldsymbol{n}_k(\boldsymbol{x}, \xi_t^{(k)})\|^2] \leq \sigma_k^2.$$

*A1* is a very generic assumption. *A2* is not assumed in the general non-convex case, but only in the $\mu$-PL cases in our theorems, instead of convexity. *A3* is implied by smoothness, and equivalent up to multiplying $\delta$ by a constant to (Karimireddy et al., 2020, Assumption A2), and appears for quadratic functions in (Shamir et al., 2014; Reddi et al., 2016; Karimireddy et al., 2019). *A4* is also very generic, and coincides with (Ajalloeian & Stich, 2020, Assumption 4). Similar assumptions to bound the bias appeared also in (Bertsekas & Tsitsiklis, 2000, though they require vanishing bias), in (Bertsekas, 2002, pg. 38–39) and more recently in (Karimireddy et al., 2020; 2019; Deng et al., 2020). *A5* can be relaxed to allow an additional unbounded variance term which grows with the norm of the estimated gradient. Convergence results under this relaxed assumption are provided in the supplementary material. Our main conclusions are maintained in this generalized case.
**Hessian dissimilarity** $\delta$: We note that Hessian dissimilarity as in *A2* for $\delta = 2L$ is directly implied by $L$-smoothness of the users. In practice, if users are similar (and not adversarial) we expect $\delta \ll L$.
**Bias parameters** $m$ **and** $\zeta^2$: To showcase the intuition behind the bias parameters $m$ and $\zeta$ we can limit ourselves to the case of one collaborator '1'. The parameter $\zeta$ quantifies translation between $f_0$ and $f_1$, while $m$ quantifies the scaling. To be more precise, if we were collaborating with a translated copy i.e. $f_1(\boldsymbol{x}) \equiv f_0(\boldsymbol{x}) + \boldsymbol{a}^\top \boldsymbol{x} + b$ then $\zeta^2 = \|\boldsymbol{a}\|^2$ and $m = 0$. If we were collaborating with a scaled copy i.e. $f_1(\boldsymbol{x}) = s f_0(\boldsymbol{x})$ then $m = (1 - s)^2$ and $\zeta = 0$. Even simpler than this, $m$ determines whether the bias is bounded or not, $m = 0$ means the bias can be bounded independently of $\boldsymbol{x}$, this should be the simplest case. The constant bias term $\zeta^2$ also quantifies how much the two collaborators' goals are different, this can be seen from the approximation $\zeta^2 \approx \|\nabla_{\boldsymbol{x}} f_1(\boldsymbol{x}_0^\star)\|^2 \propto f_1(\boldsymbol{x}_0^\star) - f_1(\boldsymbol{x}_1^\star)$, in other words how much distant are their two stationary points that they would have found by ignoring each other. In particular, $\zeta^2 = 0$ corresponds to the case of $f_0$ and $f_1$ sharing the same optimum.

## 4 Weighted Gradient Averaging

As a first basic algorithm, we here introduce *weighted gradient averaging* and analyze its convergence in the non-convex case and under the $\mu$-PL condition. We show that for the special case of collaborative mean

estimation when every user has its own distribution, we exactly recover the existing theoretical results of Grimberg et al. (2021). While recuperating analogous main ideas, our results are more general applying to any smooth stochastic optimization problem in arbitrary dimensions with multiple collaborators.

**WGA Algorithm.** As illustrated in Algorithm 2, at each time step $t$, using the current state $\boldsymbol{x}_t$, each

---

**Algorithm 2** WGA variant of Algorithm 1

---
**Require:** $\boldsymbol{x}_0$; $\eta_t$; $\alpha_t$; $\{\tau_k\}_{k=1}^N$; $T$
  ... as Algorithm 1 ...
$\nabla$ Aggregation on user 0:
  $\boldsymbol{g}(\boldsymbol{x}_t) := (1-\alpha_t)\boldsymbol{g}_0(\boldsymbol{x}_t) + \alpha_t \sum_{k=1}^N \tau_k \boldsymbol{g}_k(\boldsymbol{x}_t)$

---

collaborator $k = 1 \ldots N$ computes $\boldsymbol{g}_k(\boldsymbol{x}_t)$ an unbiased local gradient estimate of $\nabla_{\boldsymbol{x}} f_k(\boldsymbol{x}_t)$, and sends those to user 0. Then using these gradient estimates and the collaboration weights $\alpha_t \in [0,1]$, $\{\tau_i\} \in \{\tau_i \geq 0, \sum_i \tau_i = 1\}$, the main user 0 forms

$$\boldsymbol{g}(\boldsymbol{x}_t) := (1-\alpha_t)\boldsymbol{g}_0(\boldsymbol{x}_t) + \alpha_t \sum_{k=1}^N \tau_k \boldsymbol{g}_k(\boldsymbol{x}_t) \ ,$$

and performs an SGD step with the obtained gradient estimate $\boldsymbol{g}(\boldsymbol{x}_t)$, reaching the new state $\boldsymbol{x}_{t+1} = \boldsymbol{x}_t - \eta_t \boldsymbol{g}(\boldsymbol{x}_t)$.

**Remark.** WGA is SGD applied to the "modified " function $\boldsymbol{x} \mapsto (1-\alpha_t)f_0(\boldsymbol{x}) + \alpha_t \sum_{k=1}^N \tau_k f_k(\boldsymbol{x})$.

We analyze now precisely the convergence rate of Algorithm 2 under heterogeneous data across the users, in the non-convex and $\mu$-PL case in the following Theorem 4.1.

**Theorem 4.1** (Convergence of WGA). *Under Assumptions A1, A4, A5, Algorithm 2 after $T$ rounds for constant collaboration weight $\alpha_t := \alpha < 1/\sqrt{m}$, and constant step-size $\eta_t := \eta$ satisfies the following convergence bound, (where $F_t := \mathbb{E}[f_0(\boldsymbol{x}_t)] - f_0^\star$):*

***Non-convex case.*** *For $\eta = \min\left(\frac{1}{L}, \sqrt{\frac{2F_0}{L\tilde{\sigma}^2 T}}\right)$:*

$$\frac{1-\alpha^2 m}{2T}\sum_{t=0}^{T-1}\mathbb{E}\big[\|\nabla f_0(\boldsymbol{x}_t)\|^2\big] = \mathcal{O}\left(\frac{LF_0}{T} + \sqrt{\frac{LF_0\tilde{\sigma}^2(\alpha)}{T}} + \alpha^2\zeta^2\right).$$

***$\mu$-PL case.*** *If in addition A2 holds, then for the choice $\eta = \min\left(\frac{1}{L}, \frac{\log(\max(1, \frac{2\mu F_0 T}{3L\tilde{\sigma}(\alpha)^2}))}{(1-\alpha^2 m)\mu T}\right)$: $F_T =$*

$$\tilde{\mathcal{O}}\left(F_0 \exp\left(-\frac{\mu T}{L}\right) + \frac{L\tilde{\sigma}(\alpha)^2}{\mu^2 T(1-\alpha^2 m)^2} + \frac{\alpha^2\zeta^2}{\mu(1-\alpha^2 m)}\right),$$

*where $\tilde{\mathcal{O}}$ suppresses $\log(T)$ factors and we defined $\tilde{\sigma}^2(\alpha) := (1-\alpha)^2\sigma_0^2 + \alpha^2\sum_{k=1}^N \tau_k^2\sigma_k^2$ and $\zeta^2 = \sum_{k=1}^N \tau_k\zeta_k^2$.*

Similar to (Karimi et al., 2016, Theorem 4), we can get rid of the logarithmic factors in the $\mu$-PL case by choosing a decreasing step size.

**Bias-variance trade-off.** Crucially, the collaborative variance $\tilde{\sigma}^2(\alpha)$ is smaller than the individual variance $\sigma_0^2$ of user 0's gradient estimates, however, this decrease in variance is accompanied by an additional bias term $\mathcal{O}(\alpha^2\zeta^2)$, hence we have established a bias-variance trade-off, which motivates the proper choice of the collaboration weight $\alpha$.

**Choice of $\{\tau_k\}_{k=1}^N$.** The best choice of $\{\tau_k\}_{k=1}^N$ is based on a constrained quadratic programming problem (see App. C.2). However as $T \to \infty$ this best choice of the weights $\{\tau_k\}_{k=1}^N$ is completely dictated by the bias term. We have $\tau_k \propto 1_{\{k=\arg\min_l \zeta_l^2\}}$ i.e the best we can do is collaborate with the agents with the smallest bias.

**Application of WGA to collaborative mean estimation.** Weighted gradient averaging generalizes the model averaging problem studied in (Donahue & Kleinberg, 2020; Grimberg et al., 2021). We show how to recover their results here.

Suppose we want to estimate the mean $\mu_0$ of real random stochastic samples $\{z_0^{(0)}, \ldots, z_0^{(T)}\}$ with $\mathbb{E}[z_0^{(t)}] = \mu_0$. Consider

$$\min_x \ f_0(x) := \tfrac{1}{2}(x - \mu_0)^2\,,$$

with unbiased stochastic gradients given as $\nabla f(x; z_0^t) = (x - z_0^t)$. Similarly, we define our collaborator $f_1(x) := \frac{1}{2}(x - \mu_1)^2$ with a different mean $\mu_1$ and its stochastic gradients. We have that $f_0$ is 1-PL, 1-smooth, $\zeta^2 = (\mu_1 - \mu_0)^2$, and $m = 0$. Let us also use a starting point $x^0 = z_0^0$ to get $E[F_0] \le \sigma_0^2$. Plugging these values into Theorem 4.1, we get that

$$\mathbb{E}(x_T - \mu_0)^2 \le \tilde{\mathcal{O}}\left(\sigma_0^2 \exp\left(-T\right) + \frac{\tilde{\sigma}(\alpha)^2}{T} + \alpha^2(\mu_0 - \mu_1)^2\right)$$

Note that $T$ here represents the number of stochastic samples of $\mu_0$ we use. Compare this with (Grimberg et al., 2021) who show a rate of $\mathcal{O}\left(\frac{\tilde{\sigma}(\alpha)^2}{T} + \alpha^2(\mu_0 - \mu_1)^2\right)$. Thus, we recover their results for a large enough $T$ and ignoring logarithmic factors. These logarithmic factors can be avoided by using a decreasing step size (see Appendix C.2).

**Speedup over training alone.** Due to the bias-variance trade-off in Theorem 4.1, the best choice of $\alpha$ is

$$\alpha_{\mathrm{opt}} = \underset{\alpha \in (0, \frac{1}{\sqrt{m}})}{\arg\min} \ \frac{L\tilde{\sigma}(\alpha)^2}{\mu^2 T(1 - \alpha^2 m)^2} + \frac{\alpha^2 \zeta^2}{\mu(1 - \alpha^2 m)}.$$

We show that a linear speedup can only be obtained if $m = 0$ and $\zeta^2 = 0$, this means $f_k \equiv f_0$ (collaboration with $N$ copies), in this case the inverse of the speedup is given by $1 - \alpha_{\mathrm{opt}} = \frac{1}{N+1}$. However, when the functions are minimized at the same point ($\zeta^2 = 0$) but with unbounded bias ($m > 0$), the collaboration weight $\alpha$ is bounded by $\frac{1}{\sqrt{m}}$ due to the term $1 - \alpha^2 m$ in the denominator and leads to a speedup relative to training alone that is sub-linear (see Figure 6).

In the case where $\zeta^2 > 0$, the speedup gained due to weighted averaging is further limited. In fact, in this case when $T \to \infty$ we have $\alpha_{\mathrm{opt}} \to 0$ making the gain 0. Intuitively, WGA controls for the bias introduced by using gradient estimates from the collaborators by down-weighting them. While this may reduce the bias in a single round, the bias keeps accumulating over multiple rounds. Thus, the benefit of WGA diminishes with increasing $T$. In the next section, we see how to directly remove this bias.

## 5 Bias Correction

In Section 4, bias was identified as the major problem limiting the performance of WGA. Therefore we propose a bias correction algorithm that directly tackles this issue. Our strategy consists of estimating the bias between the gradients of $f_0$ and its collaborators $\{f_k\}_{k=1}^N$ using past gradients. Then, this bias is subtracted from the current gradient estimates of each collaboratorWe first demonstrate the utility of such bias correction assuming access to some ideal bias oracle. Then, we show how to use an exponential moving average of past gradients to approximate the oracle.

---

**Algorithm 3** Bias correction variant of Algorithm 1

---

**Require:** $\boldsymbol{x}_0$; $\eta_t$; $\alpha_t$; $\beta_t$; $T$; $\boldsymbol{c}_0 = \boldsymbol{b}_0$

  ... as Algorithm 1 ...

$\nabla$ Aggregation on user 0:

  $\boldsymbol{g}_{\mathrm{avg}} := \sum_{k=1}^N \tau_k \boldsymbol{g}_k(\boldsymbol{x}_t)$

  $\boldsymbol{g}(\boldsymbol{x}_t) := (1 - \alpha_t)\boldsymbol{g}_0(\boldsymbol{x}_t) + \alpha_t(\boldsymbol{g}_{\mathrm{avg}} - \boldsymbol{c}_t)$                          $\triangleright$ update

  $\boldsymbol{b}_t := \boldsymbol{g}_{\mathrm{avg}}(\boldsymbol{x}_t) - \boldsymbol{g}_0(\boldsymbol{x}_t)$                                     $\triangleright$ observed bias

  $\boldsymbol{c}_{t+1} := (1 - \beta_t)\boldsymbol{c}_t + \beta_t \boldsymbol{b}_t$                                  $\triangleright$ next bias estimate

---

**BC Algorithm.** As usual, at each time $t$, each user $k = 0, \ldots, N$ computes their own local gradient estimate $\boldsymbol{g}_k(\boldsymbol{x}_t)$. Then, as illustrated in Algorithm 3, user 0 uses $\boldsymbol{c}_t$—an estimate of the bias $c_t \approx (\sum_{k=1}^N \tau_k \nabla f_k(\boldsymbol{x}) - \nabla f_0(\boldsymbol{x}))$—and the collaboration weight $\alpha_t$:

$$\boldsymbol{g}(\boldsymbol{x}_t) := (1 - \alpha_t)\boldsymbol{g}_0(\boldsymbol{x}_t) + \alpha_t \Big( \sum_{k=1}^{N} \tau_k \boldsymbol{g}_k(\boldsymbol{x}_t) - \boldsymbol{c}_t \Big) \, .$$

Then user 0 updates their parameters using this pseudo gradient as $\boldsymbol{x}_{t+1} = \boldsymbol{x}_t - \eta_t \boldsymbol{g}(\boldsymbol{x}_t)$. We next discuss how to compute this estimate $\boldsymbol{c}_t$.

## 5.1 Using a Bias Oracle

As a warm-up, let us suppose we have access to an oracle that gives a noisy unbiased estimate of the true bias

$$\boldsymbol{c}_{\text{oracle},t} = \sum_{k=1}^{N} \tau_k \nabla_{\boldsymbol{x}} f_k(\boldsymbol{x}_t) - \nabla_{\boldsymbol{x}} f_0(\boldsymbol{x}_t) + \boldsymbol{n}_{\text{oracle},t} \, .$$

The quantity $\boldsymbol{n}_{\text{oracle},t}$ is the noise of the oracle and is independent of the gradient estimates. Using this, we have that the update satisfies

$$\mathbb{E}\Big[ \sum_{k=1}^{N} \tau_k \boldsymbol{g}_k(\boldsymbol{x}_t) - \boldsymbol{c}_{\text{oracle},t} \Big] = \nabla f_0(\boldsymbol{x}) \, .$$

Hence, this becomes similar to the case where $\zeta^2 = 0$ and $m = 0$ with WGA, enabling linear speedup. Theorem 5.1 formalizes this intuition.

**Theorem 5.1** (Convergence given a bias oracle)**.** *Under Assumption A1, using an ideal oracle of the mean bias $\boldsymbol{c}_{\text{oracle},t}$ with variance $\mathbb{E}[\|\boldsymbol{n}_{\text{oracle},t}\|^2] = v^2/N$ (i.e., $v^2$ is the variance of the bias oracle associated to each collaborator), for constant collaboration weight $\alpha_t := \alpha$, and constant step-size $\eta_t := \eta$ we have the following:*

**Non-convex case.** *For $\eta = \min\left( \frac{1}{L}, \sqrt{\frac{2F_0}{L\tilde{\sigma}^2(\alpha)}} \right)$:*

$$\frac{1}{2T} \sum_{t=0}^{T-1} \mathbb{E}[\|\nabla f_0(\boldsymbol{x}_t)\|^2] = \mathcal{O}\left( \frac{LF_0}{T} + \sqrt{\frac{LF_0\tilde{\sigma}^2(\alpha)}{T}} \right) \, .$$

*$\mu$-PL case. If in addition A2 holds, then for the choice $\eta = \min\left( \frac{1}{L}, \frac{\log(\max(1, \frac{2\mu F_0 T}{3L\tilde{\sigma}(\alpha)^2}))}{\mu T} \right)$:*

$$F_T = \tilde{\mathcal{O}}\left( F_0 \exp\left(-\frac{\mu T}{L}\right) + \frac{L\tilde{\sigma}(\alpha)^2}{\mu^2 T} \right) \, ,$$

*where $\tilde{\sigma}^2(\alpha) = (1-\alpha)^2 \sigma_0^2 + \alpha^2(\sigma_a^2 + \frac{v^2}{N})$, $\sigma_a^2 = \sum_{k=1}^{N} \tau_k^2 \sigma_k^2$.*

**Choice of the weights $\tau_k$.** We choose these weights so that we minimize $\tilde{\sigma}^2(\alpha)$, it is easy to show that there is a choice such that $\sigma_a^2 \leq \frac{\sum_{k=1}^{N} \sigma_k^2}{N^2}$. To simplify the discussion we suppose $\frac{\sum_{k=1}^{N} \sigma_k^2}{N} = \sigma_0^2$ and replace $\sigma_a^2$ by $\sigma_0^2/N$.

**Speedup over training alone.** First, note that the rate of Theorem 5.1 when $v^2 = 0$ matches Theorem 4.1 with $m = 0$ and $\zeta^2 = 0$. We examine two cases.

- If $\sigma_0^2 > 0$ : In this case, we choose

$$\alpha_{\text{opt}} \in \arg\min_{\alpha} \tilde{\sigma}^2(\alpha) = \frac{N}{N + 1 + \frac{v^2}{\sigma_0^2}} \, ,$$

giving $\tilde{\sigma}^2(\alpha_{\text{opt}}) = \sigma_0^2 \frac{1 + \frac{v^2}{\sigma_0^2}}{N + 1 + \frac{v^2}{\sigma_0^2}}$. For $N$ large enough ($N \geq \frac{v^2}{\sigma^2} + 1$), this simplifies to $\tilde{\sigma}^2(\alpha_{\text{opt}}) = O(\frac{\sigma_0^2}{N})$ and a convergence rate of $O(\sqrt{\frac{\sigma_0^2}{NT}})$ in the general non-convex case and $O(\frac{\sigma_0^2}{\mu^2 NT})$ with $\mu$-PL inequality. Thus, we achieve linear speedup.

- If $\sigma_0^2 = 0$ the baseline here is gradient descent. If $v^2 \neq 0$ then both the non-convex and $\mu$-PL convergence rates are slower than GD. The best choice of collaboration weight $\alpha$ here is $\alpha = 0$.

## 5.2 Approximating the Oracle Using EMA

Clearly, the previous discussion shows that given access to a bias oracle, using bias correction gives significant speedup even when we have a large bias i.e. $m$ and $\zeta^2$ are large. Algorithm 3 shows how we can use the exponential moving average of past gradients to estimate this bias without an oracle:

$$\boldsymbol{c}_{t+1} := (1 - \beta_t)\boldsymbol{c}_t + \beta_t\Big(\sum_{k=1}^{N} \tau_k \boldsymbol{g}_k(\boldsymbol{x}_t) - \boldsymbol{g}_0(\boldsymbol{x}_t)\Big).$$

Intuitively, this averages over $\approx \frac{1}{\beta}$ past independent stochastic bias estimates reducing the variance of $\boldsymbol{c}_t$. We next examine the effect of replacing our bias oracle using such a $\boldsymbol{c}_t$.

**Theorem 5.2** (Convergence of bias correction). *Under Assumptions A1 and A3–A5, Algorithm 3 for constant collaboration weight $\alpha_t := \alpha$, constant step-size $\eta_t := \eta \leq \min(\frac{1}{L}, \frac{1}{6\alpha^2\delta^2})$ satisfies the following:*

***Non-convex case.*** *for $\beta_t = \min(1, \left(\frac{10\delta^2(\tilde{\zeta}^2/T + \sigma_0^2 + \sigma_a^2)}{\sigma_0^2 + \sigma_a^2}\right)^{1/3}\eta^{2/3})$ we have:*

$$\frac{1}{4T}\sum_{t=0}^{T-1}\mathbb{E}[\|\nabla f_0(\boldsymbol{x}_t)\|^2] \leq \frac{F_0}{\eta T} + \frac{4\alpha^2 E_0}{\beta T} + 12\alpha^2\left((\sigma_0^2 + \sigma_a^2)(\tilde{\zeta}^2/T + \sigma_0^2 + \sigma_a^2)\right)^{1/3}(\delta\eta)^{2/3} + \frac{L\sigma^2(\alpha)}{2}\eta + 10\alpha^2\delta^2\sigma^2(\alpha)\eta^2.$$

***$\mu$-PL case.*** *for $\beta_t = \min(1, \left(10\delta^2\right)^{1/3}\eta^{2/3})$ we have:*

$$F_T \leq (1 - \frac{\mu\eta}{2})^T\Phi_0 + \frac{L\sigma^2(\alpha)}{\mu}\eta + 24\alpha^2\left(\sigma_0^2 + \sigma_a^2\right)^{2/3}(\delta\eta)^{2/3}/\mu.$$

*where $\Phi_0 = F_0 + \frac{2\alpha^2\eta}{\beta}E_c^0 + \frac{10\alpha^2\eta}{\beta^2}\tilde{\zeta}$, $F_t = \mathbb{E}[f_0(\boldsymbol{x}_t)] - f_0^\star$, $E_0 = \mathbb{E}[\|\boldsymbol{c}_0 - \nabla f_1(\boldsymbol{x}_0) + \nabla f_0(\boldsymbol{x}_0)\|^2], \sigma^2(\alpha) := (1 - \alpha)^2\sigma_0^2 + \alpha^2\sigma_a^2$, $\tilde{\zeta}^2 := 2(1 + m)(\mathbb{E}[\|\nabla f_0(\boldsymbol{x}_0)\|^2] + 2\zeta^2)$, $\sigma_a^2 = \sum_{k=1}^{N}\tau_k^2\sigma_k^2$ and $\zeta^2 = \sum_{k=1}^{N}\tau_k\zeta_k^2$.*

**Discussion:**

- **Significance of the terms.** In the non-convex inequality the first term in the inequality in theorem 5.2 measures how fast the initial condition is forgotten, the second term measures how the initial bias estimation affects the optimization whereas the third term measures the effect of having used noisy (and dependent on the past) estimates of the bias.

- **Bias correction works** We see that $\zeta^2$ is divided by $T$ which means that our bias correction strategy works indeed in correcting the bias $\zeta^2$. However, using EMA adds the term $12\alpha^2\big((\sigma_0^2 + \sigma_a^2)(\tilde{\zeta}^2/T + \sigma_0^2 + \sigma_a^2)\big)^{1/3}(\delta\eta)^{2/3}$ which is greater than the noise term $\frac{L\sigma^2(\alpha)}{2}\eta$ unless we limit ourselves to collaborators with small Hessian dissimilarity $\delta$.

- **Condition on the dissimilarity $\delta$.** Theorem 5.2 shows that to gain from the collaboration (be better than training alone) we need $(\delta\eta)^{2/3} \ll \eta$. Now if we fix $T$, The optimal $\eta$ in the non-convex case is of order $\frac{1}{\sqrt{T}}$ , then we would need $\delta^2 = o(\frac{1}{\sqrt{T}})$ and in the $\mu$-PL case the optimal $\eta$ scales as $\frac{1}{T}$ so that we need $\delta^2 = o(\frac{1}{T})$ in this case .

**Remark.** The condition on the similarity parameter $\delta$ is reasonable since, in particular, it eliminates adversarial agents that would have a big $\delta$.

**Choice of the weights $\tau_k$.** We show (See C.4) that as $T \to \infty$ the best choice of these weights is completely dictated by the variance term $\sigma_a^2$. In particular there is always a choice such that $\sigma_a^2 \leq \frac{\sum_{k=1}^{N}\sigma_k^2}{N^2}$. This means that $\sigma_a^2$ scales as $1/N$ so for simplification's sake we suppose $\frac{\sum_{k=1}^{N}\sigma_k^2}{N} \leq \sigma_0^2$ and replace $\sigma_a^2$ by $\sigma_0^2/N$.

**Corollary 5.3** (linear speedup of BC). *For $\sigma_0^2 > 0$ and a fixed horizon $T$, supposing that we have a mechanism to select collaborators with $\delta^2 = o(\frac{1}{\sqrt{T}})$ in the non-convex case and $\delta^2 = o(\frac{1}{T})$ in the $\mu$-PL case. Then there is an appropriate choice of the weights $\alpha, \{\tau_k\}$ for which, in leading order of $T$ we have :*

***Non-convex case.*** *For $\eta = \min(1/L, 1/(6\alpha^2\delta^2), \sqrt{\frac{2F_0}{L\sigma^2(\alpha)T}})$:*

$$\frac{1}{T}\sum_{t=0}^{T-1}\mathbb{E}[\|\nabla f_0(\boldsymbol{x}_t)\|^2] = \mathcal{O}\left(\sqrt{\frac{LF_0\sigma_0^2}{(N+1)T}}\right).$$

***$\mu$-PL case.*** *for $\eta = \min(1/L, 1/(6\alpha^2\delta^2), \frac{\log(\max(2,\frac{2\mu\Phi_0 T}{3L\sigma^2(\alpha)}))}{\mu T})$ :*

$$F_T = \tilde{\mathcal{O}}\left(\Phi_0\frac{L\sigma_0^2}{\mu^2(N+1)T}\right)$$

**Remark.** It is not hard to see that the quantities $\zeta$ and $\delta$ are "perpendicular" in the sense that $\delta$ can be small and $\zeta$ very big. For example, we can take $f_0(x) = \frac{1}{2}x^2$ and $f_1(x) = \frac{1+\delta}{2}(x - \frac{\zeta}{1+\delta})^2$. Corollary 5.3 means that we can benefit optimally from all agents that have a small $\delta$ irrespective of their bias $\zeta^2$.

**Conclusion: BC solves "partially" the problems of WGA.** From the above discussion, we see that BC solves the problems WGA had with bias parameters $m$ and $\zeta^2$. First of all, there is no dependence on the heterogeneity parameter $m$, in particular the collaboration weight $\alpha$ can range freely in the interval $[0, 1]$. Secondly, with BC, the bias $\zeta^2$ does not accumulate with time. However, we only benefit optimally from our EMA approach when the dissimilarity $\delta$ between the collaborators is small $\delta^2 = o(\frac{1}{\sqrt{T}})$ (No Free Lunch).

## 6 Experiments

To validate our theory we consider the noisy quadratic model i.e. optimizing a function of the type

$$f_0(\boldsymbol{x}) := \tfrac{1}{2}(\boldsymbol{x} - \boldsymbol{m}_0^\star)^\top \boldsymbol{A}_0(\boldsymbol{x} - \boldsymbol{m}_0^\star), \ \boldsymbol{m}_0^\star \sim \mathcal{N}(\boldsymbol{x}_0^\star, \boldsymbol{\Sigma}_0) .$$

While simple, this model can serve as an illustrative test for our theory and is often used to test machine learning and federated learning algorithms (Schaul et al., 2013; Wu et al., 2018; Martens & Grosse, 2015; Zhang et al., 2019). One common simplification is to consider both $\boldsymbol{A}_0$ and $\boldsymbol{\Sigma}_0$ to be diagonal (or co-diagonalizable). This assumption makes it possible to optimize the function $f_0$ over each of its dimensions independently. So it suffices to consider a noisy quadratic model in 1D: optimizing $f_0(x) := \frac{1}{2}a_0(x - x_0^\star + \frac{\xi_0}{a_0})^2, \xi_0 \sim \mathcal{N}(0, \sigma^2)$ by collaborating with $f_{avg}(x) := \frac{1}{2}a_1(x - x_1^\star + \frac{\xi_1}{a_1})^2, \xi_1 \sim \mathcal{N}(0, \frac{\sigma^2}{N})$. Here, we have $N$ as the number of collaborators, $\delta = \|a_0 - a_1\|$, and $\zeta^2 = \|a_1(x_1^\star - x_0^\star)\|^2$. The quantity $f_{0,test} = \frac{1}{2}a_0(x - x_0^\star)^2$ can be interpreted as a test loss (called simply loss in the plots). In our plots we use by default $\delta = 1, \zeta = 4$ and $\sigma = 10$.

**Convergence speed.** Figure 1 shows convergence curves of the three competing algorithms we have discussed before: working alone, weighted gradient averaging (WGA), and bias correction (BC). In particular, we see that BC reaches a lower error level compared to both other algorithms. This confirms our theory that BC reduces the bias in the algorithm enabling it to reach a lower error level. The initial increase in the loss is also characteristic of BC and is because during the initial stages our EMA estimate of the bias is quite poor. Eventually, the bias estimate improves and we get fast convergence.

**Dependence on data heterogeneity.** Figure 2 shows how the bias parameter $\zeta^2$ influences the performance of BC. As predicted by the theory, we see that BC always converges to the same error level uninfluenced by $\zeta^2$. This bias only effects the time horizon needed for convergence. In contrast, WGA is strongly influenced by the bias as we see in Figure 3. In fact, the convergence error level of WGA is directly proportional to $\alpha^2\zeta^2$, meaning that we would need to set $\alpha = 0$ (i.e train alone) to ensure low error. This demonstrates that the bias correcting technique employed by BC indeed succeeds, validating our theory.

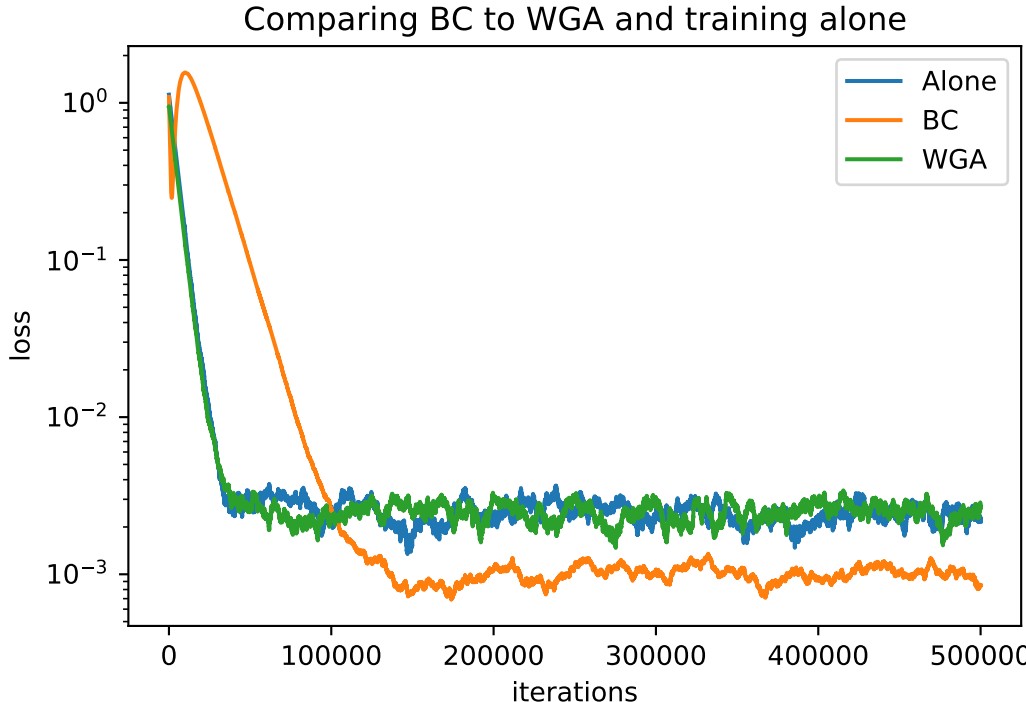

Figure 1: Comparing Bias Correction (orange) to WGA (green) and training alone (blue). BC achieves a lower loss ~~than training alone or using WGA. There are noise process~~ for training alone and WGA but not for BC

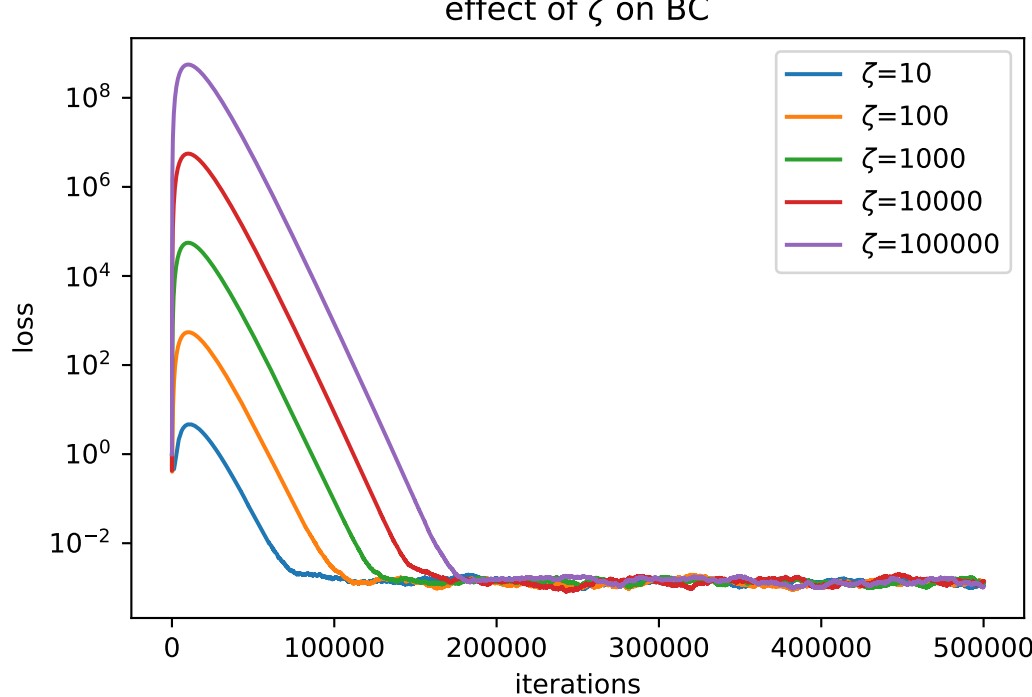

Figure 2: Effect of the bias $\zeta$ on the convergence of BC for a fixed choice of step-size $\eta = 10^{-4}$, BC weight $\beta = 10^{-4}$ and collaboration weight $\alpha = \frac{N}{N+1}$, where $N = 10$. We can see that $\zeta$ influences the time needed for convergence but eventually all curves converge to the same error level.

**Dependence on the number of collaborators.** Figure 4 shows how the number of collaborators $N$ influences the convergence of BC for a relatively big $\delta = 1$. We see that increasing $N$ does have a positive effect on BC and decreases the error level to which it converges. However, the benefit saturates quickly. While there is a substantial improvement from $N = 1$ to $N = 10$, the rest only sees negligible improvement.

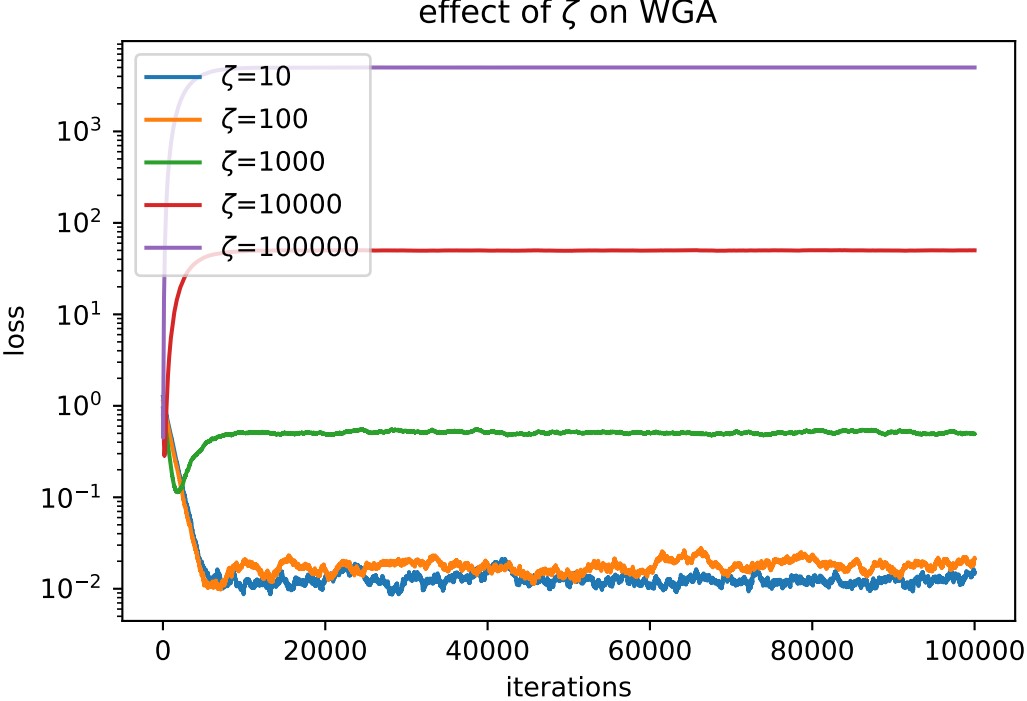

Figure 3: Effect of the bias $\zeta$ on the convergence of WGA for a fixed choice of step-size $\eta = 5 \times 10^{-4}$, collaboration weight $\alpha = 10^{-3}$ and $N = 10$. We can see that the bigger $\zeta$ is the bigger the final loss will be. In fact, WGA can only converge up to $\mathcal{O}(\alpha^2 \zeta^2)$.

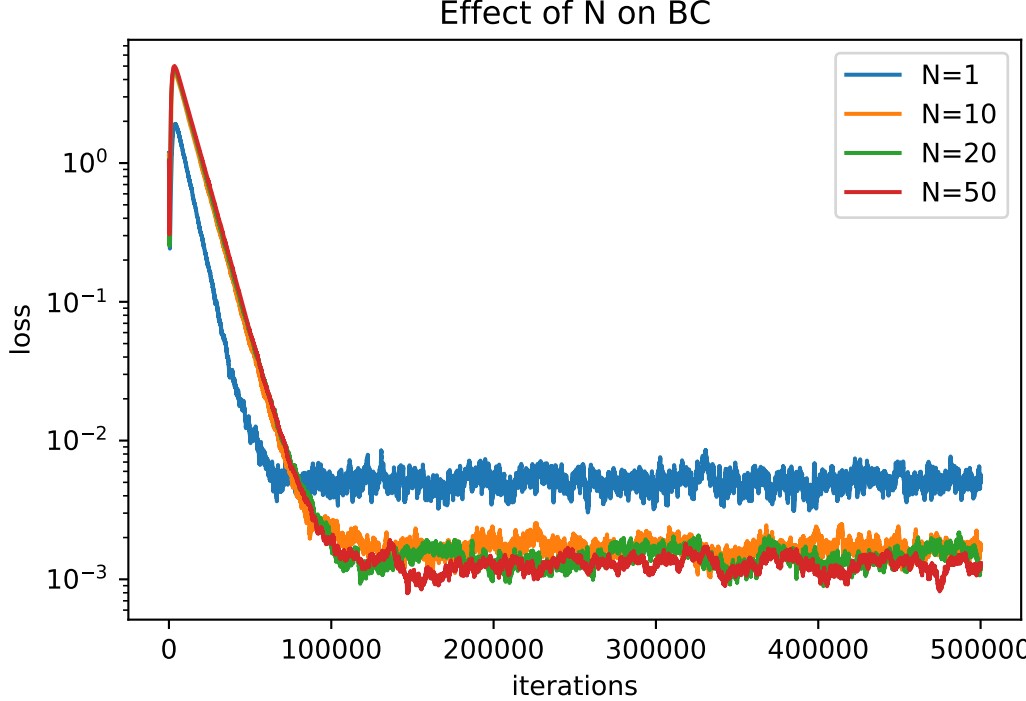

Figure 4: Effect of the number of collaborators $N$ on the convergence of BC for a fixed choice of step-size $\eta = 5 \times 10^{-4}$, BC weight $\beta = 10^{-4}$, collaboration weights $\alpha = \frac{N}{N+1}$ and $\delta = 1$ (not very small). We can see that increasing $N$ does improve the level to which BC converges, due to the smaller variance with larger $N$.

We expect this to result from using a big $\delta$ since our theory only predicts linear speedup in $N$ for $\delta$ very small.

## 7 Limitations and Extensions

**Bias and generalization.** We have proposed a strategy to correct for the "gradient" bias between the main agent and its collaborators, but in doing so we have put a lot of faith in the "quality" of the main agent's gradients. However, in the case where the main agent has a very limited dataset, some bias might be good to make out for the lack of data.

**Bias Correction in deep learning.** In this work we have employed the idea of gradient bias correction using SGD. Our methods can also be extended to other optimizers such as momentum or Adam. A larger empirical exploration of such algorithms, as well as more real-world deep learning experiments, would be valuable but is out of scope for our more theoretical work.

**Adding local steps.** Currently, the users communicate with each other after every gradient computation. This is a problem for Federated Learning (which is not the aim of this paper). More communication-efficient schemes can be developed by instead allowing multiple local steps before communication, such as in FedAvg (McMahan et al., 2017). Similarly, extending our algorithms to allow personalization for all users instead of focusing only on user 0 would improve practicality in the federated learning setting.

**Fine-grained measures of similarity.** Our algorithms, as well as the assumptions, use static global measures of dissimilarity. Time-varying adaptive weighting strategies such as cosine similarity between gradients may further improve our algorithms. Using individual user-level similarities, such as in (Grimberg et al., 2020) would also be a fruitful extension. Similarity-based user selection rules are also closely related to Byzantine robust learning, where they are used to exclude malicious participants (Blanchard et al., 2017; Baruch et al., 2019; Karimireddy et al., 2021).

## 8 Conclusion

In this work, we have introduced the collaborative stochastic optimization framework where one "main" user collaborates with a set of willing-to-help collaborators. We considered the simplest method to solve this problem: using SGD with *weighted gradient averaging*. We discussed in detail the limitations of this idea arising mainly due to the bias introduced by the collaboration. To solve this bias problem, we proposed a second algorithm *bias correction*. We showed that our bias correction algorithm manages to remove the effect of this bias and, under some optimal choices of its parameters, leads to a linear speedup as we increase the number of collaborators.

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

# A More related work and discussion

## A.1 Related work

In personalized Federated Learning, a prominent approach consists in using a local-global interpolation such as in (Hanzely & Richtárik, 2020) which proposes to use a consensus-like regularization to make such an interpolation, however, they only give convergence of the global model. In a second work (Hanzely et al., 2021) study the problem of optimizing an objective that has both local and global parameters, they propose an SVRG-like algorithm to reduce the variance of local gradient estimates. We reiterate that our goal differs from that of Federated Learning, we care about the performance of one particular agent, and the bias we have is inherent to collaborating with different agents, it is not a result of using local steps as in Federated Learning. Also, our bias correction method has the main goal to reduce the bias, the reduction in variance is a result of averaging and further using an exponential moving average to reduce the variance of our bias estimates. (Li et al., 2020b) discusses variance trade-offs for point estimation and linear regression problems, our results are more general from this perspective.

In the personalized optimization setting, two very recent empirical works propose rules to learn collaboration weights. Beaussart et al. (2021) modify Scaffold (Karimireddy et al., 2019) to use Euclidean distances of the updates between different agents to derive a heuristic for weight definition. It uses both local and global control variates, though without a decay mechanism. (Zhang et al., 2021) on the other hand uses an idea from meta-learning to learn the collaboration weights, by using a first-order approximation of the objective with respect to these weights. While demonstrating practical performance on deep learning tasks, neither of the two methods comes with convergence guarantees. Our approach in contrast chooses collaboration weights to achieve provable convergence as well as speedup with the number of workers.

## A.2 Comparison with other control variate techniques

Control variates have been used extensively in variance reduction techniques such as SVRG (Johnson & Zhang, 2013), SAGA (Defazio et al., 2014), SAG (Schmidt et al., 2013), MVR (Cutkosky & Orabona, 2019). The main idea is the following: given an unbiased gradient estimate $g(\boldsymbol{x})$ at $\boldsymbol{x}$, to reduce its variance we replace $g(\boldsymbol{x})$ by $g(\boldsymbol{x}) - \boldsymbol{y} + E[\boldsymbol{y}]$ where $\boldsymbol{y}$ is a random variable that correlates positively with $g(\boldsymbol{x})$, this is true for all the methods cited before except for SAG which does not bother to keep the new gradient estimate unbiased. This idea is used in Federated Learning to correct for the bias introduced by the use of local steps, In SCAFFOLD (Karimireddy et al., 2019) for example, $g(\boldsymbol{x})$ is the $i$th client gradient estimate at $\boldsymbol{x}$ the current local model, and $\boldsymbol{y}$ is the gradient estimate of the same client but at the last received server model $\boldsymbol{z}$ and $E[\boldsymbol{y}]$ is client average true gradient at $\boldsymbol{z}$. Thus the bias of this new gradient estimate is $\nabla f_i(\boldsymbol{x}) - \nabla f(\boldsymbol{x}) - \nabla f(\boldsymbol{z}) + \nabla f(\boldsymbol{z})$, by assuming $f_i - f$ has a hessian bounded by $\delta$ it is easy to see that the norm of the bias will be bounded by $\delta \|\boldsymbol{x} - \boldsymbol{z}\|$, all that is left is to efficiently bound this norm $\|\boldsymbol{x} - \boldsymbol{z}\|$.

In our case, the bias does not come from local steps but is a result of collaborating with potentially different agents. We note here that our goal is different from that of Federated Learning which aims to train the average model, whereas we train one model using/collaborating with other models and we only care about local performance. Again, the bias is inherent to the collaboration, our solution to reduce it is estimating the future bias based on past observed biases and then subtracting it from the current gradient estimate.

For simplicity, let's discuss the case $\beta = 1$ which means we only use the last observed bias to estimate the current bias. In this case, the bias of our corrected gradient is : $\alpha(\nabla f_1(\boldsymbol{x}_t) - \nabla f_0(\boldsymbol{x}_t) - \nabla f_1(\boldsymbol{x}_{t-1}) + \nabla f_0(\boldsymbol{x}_{t-1}))$ , using the bounded hessian dissimilarity assumption, it is easy to see that the norm of this quantity is bounded by $\alpha\delta\|\boldsymbol{x}_t - \boldsymbol{x}_{t-1}\|$, if we can efficiently bound this quantity, the convergence proof will be easy. It turned out that using only the last observed bias as a bias estimate incurs the same additional variance in all steps, to solve this we propose the use of an exponential average of all past observed biases.

One important idea about these approaches that use bounded hessian dissimilarity and lead to bounding the bias as above is that this makes it possible to control the bias of the model indirectly by controlling the step size.

## B Relaxing Noise Assumptions

We start by relaxing our assumptions about the noise. In general, we can make the following assumptions:

*First relaxation of A5 (Bounded variance)* for each agent $k \in \{0, \ldots, N\} \ \exists \ M_k, \sigma_k^2 \geq 0$ s.t. $\forall \boldsymbol{x} \in \mathbb{R}^d$:

$$\left\{ \ \mathbb{E}[\|\boldsymbol{n}_k(\boldsymbol{x}, \xi_t^{(k)})\|^2] \leq M_k \|\nabla_{\boldsymbol{x}} f_k(\boldsymbol{x})\|^2 + \sigma_k^2 \, . \right.$$

The quantity $\sigma_k^2$ is the variance of collaborator's gradient estimates when agent $k$ has converged to a stationary point. Using this new assumption with the gradient dissimilarity assumption:
*A4 (Gradient Similarity)* $\exists \ m, \zeta_k^2 \geq 0$ s.t. $\forall \boldsymbol{x} \in \mathbb{R}^d$:

$$\|\nabla_{\boldsymbol{x}} f_k(\boldsymbol{x}) - \nabla_{\boldsymbol{x}} f_0(\boldsymbol{x})\|^2 \leq m \|\nabla_{\boldsymbol{x}} f_0(\boldsymbol{x})\|^2 + \zeta_k^2 \, .$$

Now if we denote $f_{\text{avg}} = \sum_{k=1}^N \tau_k f_k$ and $\boldsymbol{n}_{\text{avg}}$ the variance associated to its gradient estimate, we have :

$$\left\{ \begin{array}{ll} \mathbb{E}[\|\boldsymbol{n}_{\text{avg}}(\boldsymbol{x}, \xi_t^{i=(1\ldots N)})\|^2] \leq M_{\text{avg}} m' \|\nabla_{\boldsymbol{x}} f_0(\boldsymbol{x})\|^2 / N + \tilde{\sigma}_{\text{avg}}^2 / N \, , \\ \tilde{\sigma}_{\text{avg}}^2 & = N \sum_{k=1}^N \tau_k^2 \tilde{\sigma}_k^2 \, , \\ \tilde{\sigma}_k^2 & = \sigma_k^2 + 2 M_k \zeta_k^2 \, , \\ M_{\text{avg}} & = 2N \sum_{k=1}^N \tau_k^2 M_k \, , \\ m' & = 2(1+m) \, . \end{array} \right.$$

The quantity $\tilde{\sigma}_{\text{avg}}^2$ measures the average variance of collaborators' gradient estimates this time when agent "0" has converged to a stationary point. $M_k \zeta_k^2$ is the variance resulting from collaborator $k$ being biased from agent 0 and thus converging to a different minimizer. We can argue that when the hessian dissimilarity parameter $\delta = 0$ i.e. each collaborator $f_k$ is a translated copy of $f_0$ then the noise will not be changed from its original level by translation (adding a constant to a random variable does not change its variance) and thus $M_k \zeta_k^2$ should be replaced by a quantity that is proportional to the parameter $\delta$. This motivates the final form of our assumption:

*Final form of A5 (Bounded variance)* $\exists \ \sigma_k^2, D_k^2 \geq 0$ s.t. $\forall \boldsymbol{x} \in \mathbb{R}^d$:

$$\left\{ \begin{array}{ll} \mathbb{E}[\|\boldsymbol{n}_{\text{avg}}(\boldsymbol{x}, \xi_t^{i=(1\ldots N)})\|^2] \leq M_{\text{avg}} m' \|\nabla_{\boldsymbol{x}} f_0(\boldsymbol{x})\|^2 / N + \tilde{\sigma}_{\text{avg}}^2 / N \, , \\ \tilde{\sigma}_{\text{avg}}^2 & = N \sum_{k=1}^N \tau_k^2 \tilde{\sigma}_k^2 \, , \\ \tilde{\sigma}_k^2 & = \sigma_k^2 + 2 \delta M_k D_k^2 \, , \\ M_{\text{avg}} & = 2N \sum_{k=1}^N \tau_k^2 M_k \, , \\ m' & = 2(1+m) \, . \end{array} \right.$$

$D_k^2$ is a constant that can be interpreted as a diameter of the parameter space for agent $k$.

We note that we can still safely go to the other forms of this assumption without affecting the proofs, we can always replace $\delta D_k^2$ by $\zeta_k^2$ in our next result if the reader is not convinced by the dependence of the noise with respect to $\delta$, and we can replace $\tilde{\sigma}_{\text{avg}}^2$ by $\sigma_{\text{avg}}^2 = N \sum_{k=1}^N \tau_k^2 \sigma_k^2$ if we don't want to make the noise of the collaborators when agent "0" has converged depend on their bias.

We will do the proofs for only $N = 1$ and without taking into account the dependence of the noise of agent "1" on its bias with respect to "0". To be explicit, for a collaboration with one agent "1" we make the following assumption on the noise:

$$\left\{ \begin{array}{ll} \mathbb{E}[\|\boldsymbol{n}_0(\boldsymbol{x}, \xi_t^{(0)})\|^2] & \leq M_0 \|\nabla_{\boldsymbol{x}} f_0(\boldsymbol{x})\|^2 + \sigma_0^2 \, , \\ \mathbb{E}[\|\boldsymbol{n}_1(\boldsymbol{x}, \xi_t^{i=(1)})\|^2] \leq M_1 m \|\nabla_{\boldsymbol{x}} f_0(\boldsymbol{x})\|^2 + \sigma_1^2 \, . \end{array} \right.$$

This will not make us lose any generality since we can replace $M_1$ by $M_{\text{avg}}/N$ and $\sigma_1^2$ by $\sigma_{\text{avg}}^2/N$ or $\tilde{\sigma}_{\text{avg}}^2/N$. Furthermore, we would also need to replace $\zeta^2$ by $\sum_{k=1}^N \tau_k \zeta_k^2$.

# C  Missing Proofs

## C.1  SGD with biased gradients

If we are optimizing an $L$-smooth function $f_0$ on $\mathbb{R}^d$ using SGD iterations $\boldsymbol{x}_{t+1} = \boldsymbol{x}_t - \eta_t \boldsymbol{g}(\boldsymbol{x}_t)$ with a gradient that can be written in the form

$$\boldsymbol{g}(\boldsymbol{x}_t) = \nabla_{\boldsymbol{x}} f_0(\boldsymbol{x}_t) + \underbrace{\boldsymbol{b}(\boldsymbol{x}_t)}_{bias} + \underbrace{\boldsymbol{n}_t}_{noise}$$

Then denoting $F_t = \mathbb{E}[f_0(\boldsymbol{x}_t)] - f_0^\star$, we have for $\eta_t \leq 1/L$:

$$F_{t+1} - F_t \leq \frac{\eta}{2}(-\|\nabla_{\boldsymbol{x}} f_0(\boldsymbol{x}_t)\|^2 + \|\boldsymbol{b}(\boldsymbol{x}_t)\|^2) + + \frac{L\eta^2}{2}\mathbb{E}[\|\boldsymbol{n}_t\|^2] \tag{3}$$

*Proof.* Using the $L$-smoothness of $f_0$ we have:

$$
\begin{aligned}
\mathbb{E}[f_0(\boldsymbol{x}_{t+1})] - f_0(\boldsymbol{x}_t) &\leq \langle \nabla_{\boldsymbol{x}} f_0(\boldsymbol{x}_t), \mathbb{E}[\boldsymbol{x}_{t+1} - \boldsymbol{x}_t]\rangle + \frac{L}{2}\mathbb{E}_{\xi_t}[\|\boldsymbol{x}_{t+1} - \boldsymbol{x}_t\|^2] \\
&= -\eta\langle \nabla_{\boldsymbol{x}} f_0(\boldsymbol{x}_t), \mathbb{E}[\boldsymbol{g}(\boldsymbol{x}_t)]\rangle + \frac{L}{2}\eta^2 \mathbb{E}_{\xi_t}[\|\boldsymbol{g}(\boldsymbol{x}_t)\|^2] \\
&= -\eta\langle \nabla_{\boldsymbol{x}} f_0(\boldsymbol{x}_t), \nabla_{\boldsymbol{x}} f_0(\boldsymbol{x}_t) + \boldsymbol{b}(\boldsymbol{x}_t)\rangle + \frac{L}{2}\eta^2(\|(\nabla_{\boldsymbol{x}} f_0(\boldsymbol{x}_t) + \boldsymbol{b}(\boldsymbol{x}_t)\|^2 + \mathbb{E}[\|\boldsymbol{n}_t\|^2])
\end{aligned}
$$

Using $L\eta \leq 1$ :

$$
\begin{aligned}
\mathbb{E}[f_0(\boldsymbol{x}_{t+1})] - f_0(\boldsymbol{x}_t) &\leq \frac{\eta}{2}(-2\langle \nabla_{\boldsymbol{x}} f_0(\boldsymbol{x}_t), \nabla_{\boldsymbol{x}} f_0(\boldsymbol{x}_t) + \boldsymbol{b}(\boldsymbol{x}_t)\rangle + \|\nabla_{\boldsymbol{x}} f_0(\boldsymbol{x}_t) + \boldsymbol{b}(\boldsymbol{x}_t)\|^2) + \frac{L\eta^2}{2}\mathbb{E}[\|\boldsymbol{n}_t\|^2]) \\
&= \frac{\eta}{2}(-[\|\nabla_{\boldsymbol{x}} f_0(\boldsymbol{x}_t)\|^2 + \|\boldsymbol{b}(\boldsymbol{x}_t)\|^2) + \frac{L\eta^2}{2}\mathbb{E}[\|\boldsymbol{n}_t\|^2]
\end{aligned}
$$

Taking an overall expectation yields the desired result. □

All of the proofs will use this inequality as a starting point.

## C.2  Proof of Theorem 4.1

In this section, we present the detailed proof of Theorem 1 i.e the convergence of WGA for both the non-convex and $\mu$-PL case.

We denote $n(\boldsymbol{x}, \xi_t) = (1-\alpha)\boldsymbol{n}(\boldsymbol{x}, \xi_t^{(0)}) + \alpha \boldsymbol{n}_1(\boldsymbol{x}, \xi_t^{(1)})$ the noise of the weighted average.

**Bounding the average noise.** Using Assumption $A5$ (Bounded noise), we can bound the noise in the following way:

$$E_\xi[\|\boldsymbol{n}(\boldsymbol{x}, \xi_t)\|^2] \leq M(\alpha)\|\nabla_{\boldsymbol{x}} f_0(x)\|^2 + \tilde{\sigma}^2(\alpha),$$

Where $\tilde{\sigma}^2(\alpha) := (1-\alpha)^2\sigma_0^2 + \alpha^2\sigma_1^2$ and $M(\alpha) := (1-\alpha)^2 M_0 + \alpha^2 M_1 m \leq M = M_0 + M_1 m$.

*Proof.*

$$
\begin{aligned}
E_\xi[\|n(x, \xi_t)\|^2] &= (1-\alpha)^2 E_{\xi_t^{(0)}}[\|\boldsymbol{n}_0(x, \xi_t^{(0)})\|^2] + \alpha^2 E_{\xi_t^{(1)}}[\|\boldsymbol{n}_1(x, \xi_t^{(1)})\|^2] \\
&\leq (1-\alpha)^2\{M_0\|\nabla_{\boldsymbol{x}} f_0(x)\|^2 + \sigma_0^2\} + \alpha^2\{M_1 m\|\nabla_{\boldsymbol{x}} f_0(x)\|^2 + \sigma_1^2\} \\
&\leq M(\alpha)\|\nabla_{\boldsymbol{x}} f_0(x)\|^2 + \tilde{\sigma}^2(\alpha).
\end{aligned}
$$

□

**Main inequality.** Now denoting $F_t = \mathbb{E}[f_0(\boldsymbol{x}_t)] - f_0^\star$, for $\eta \leq 1/L$, we have :

$$F_{t+1} - F_t \leq \frac{\eta}{2}(-1 + \alpha^2 m + LM\eta)\mathbb{E}\Big[\|\nabla_{\boldsymbol{x}} f_0(\boldsymbol{x}_t)\|^2\Big] + \frac{\eta\alpha^2}{2}\zeta^2 + \frac{L\eta^2}{2}\tilde{\sigma}^2(\alpha)$$

*Proof.* With $L$-smoothness of $f_0$ and $\eta L \leq 1$ we can use (3) with $\boldsymbol{b}(\boldsymbol{x}_t) = \alpha(\nabla_{\boldsymbol{x}} f_1(\boldsymbol{x}_t) - \nabla_{\boldsymbol{x}} f_0(\boldsymbol{x}_t))$ and $\boldsymbol{n}_t = \boldsymbol{n}(x, \xi_t)$

the Bounded Gradient Dissimilarity assumption $(A4)$ lets us upper-bound the term $\|\boldsymbol{b}(\boldsymbol{x}_t)\|^2 \leq \alpha^2(m\|\nabla_{\boldsymbol{x}} f_0(\boldsymbol{x})\|^2 + \zeta^2)$.

$$\mathbb{E}_{\xi_t}[f_0(\boldsymbol{x}_{t+1})] - f_0(\boldsymbol{x}_t) \leq \frac{\eta}{2}(-\|\nabla_{\boldsymbol{x}} f_0(\boldsymbol{x}_t)\|^2 + \alpha^2\|\nabla_{\boldsymbol{x}} f_0(\boldsymbol{x}_t) - \nabla_{\boldsymbol{x}} f_1(\boldsymbol{x}_t)\|^2) + \frac{L\eta^2}{2}(M\|\nabla_{\boldsymbol{x}} f_0(\boldsymbol{x})\|^2 + \tilde{\sigma}^2(\alpha))$$

$$\leq \frac{\eta}{2}(-1 + \alpha^2 m + LM\eta)\|\nabla_{\boldsymbol{x}} f_0(\boldsymbol{x}_t)\|^2 + \frac{\eta\alpha^2}{2}\zeta^2 + \frac{L\eta^2}{2}\tilde{\sigma}^2(\alpha)$$

All that is left is to take an overall expectation. $\qquad\square$

Now if $M = M_0 + mM_1 \neq 0$, then we choose $\eta \leq \frac{1 - \alpha^2 m}{2LM}$ which gives

$$F_{t+1} - F_t \leq -\frac{\eta}{4}(1 - \alpha^2 m)\|\nabla_{\boldsymbol{x}} f_0(\boldsymbol{x}_t)\|^2 + \frac{\eta\alpha^2}{2}\zeta^2 + \frac{L\eta^2}{2}\tilde{\sigma}^2(\alpha)$$

And if $M = 0$, then we get

$$F_{t+1} - F_t \leq -\frac{\eta}{2}(1 - \alpha^2 m)\|\nabla_{\boldsymbol{x}} f_0(\boldsymbol{x}_t)\|^2 + \frac{\eta\alpha^2}{2}\zeta^2 + \frac{L\eta^2}{2}\tilde{\sigma}^2(\alpha)$$

We combine these two inequalities into one:

$$F_{t+1} - F_t \leq -\frac{\eta}{c}(1 - \alpha^2 m)\|\nabla_{\boldsymbol{x}} f_0(\boldsymbol{x}_t)\|^2 + \frac{\eta\alpha^2}{2}\zeta^2 + \frac{L\eta^2}{2}\tilde{\sigma}^2(\alpha) \tag{4}$$

The constant $c$ is equal to 2 if $M = 0$ and equal to 4 otherwise. This constant is not very important since we can always choose the step-size $\eta$ small enough to make $c$ close to 1.

**Remark.** We need $1 - \alpha^2 m >= 0$ i.e. $\alpha \leq 1/\sqrt{m}$ if this bound is to guarantee any convergence.

**Non-convex case of Theorem 4.1.** To prove the non-convex result, it suffices to rearrange the terms in (4), sum for $t = 0$ to $t = T - 1$ and divide by $T$. This manipulation gives:

$$\frac{(1 - \alpha^2 m)}{cT}\sum_{t=0}^{T-1}\mathbb{E}\big[\|\nabla f_0(\boldsymbol{x}_t)\|^2\big] \leq \frac{1}{\eta T}\sum_{t=0}^{T-1}(F_t - F_{t+1}) + \frac{L\eta}{2}\tilde{\sigma}^2(\alpha) + \frac{1}{2}\alpha^2\zeta^2$$

$$\leq \frac{F_0}{\eta T} + \frac{L\eta}{2}\tilde{\sigma}^2(\alpha) + \frac{1}{2}\alpha^2\zeta^2$$

This is true for all $\eta \leq \eta_{\max} := \min\left(\frac{1}{L}, \frac{1 - \alpha^2 m}{2LM}\right)$.

Choosing $\eta = \min\left(\eta_{\max}, \sqrt{\frac{2F_0}{L\tilde{\sigma}^2 T}}\right)$ leads to the following result:

$$\frac{1 - \alpha^2 m}{cT}\sum_{t=0}^{T-1}\mathbb{E}\big[\|\nabla f_0(\boldsymbol{x}_t)\|^2\big] \leq \frac{F_0}{\eta_{\max} T} + \sqrt{\frac{2LF_0\tilde{\sigma}^2(\alpha)}{T}} + \frac{1}{2}\alpha^2\zeta^2.$$

**$\mu$-PL case of Theorem 4.1.** To prove the $\mu$-PL result, we start from (4), we use Assumption $A2$ i.e. $f_0$ satisfies the $\mu$-PL condition: $\forall \boldsymbol{x} \in \mathbb{R}^d$ $\|\nabla_{\boldsymbol{x}} f_0(\boldsymbol{x}_t)\|^2 \geq 2\mu(f_0(\boldsymbol{x}) - f_0^\star)$, this yields:

$$F_{t+1} \leq \left(1 - \frac{2\mu\eta}{c}(1 - \alpha^2 m)\right)F_t + \frac{\eta\alpha^2}{2}\zeta^2 + \frac{L\eta^2}{2}\tilde{\sigma}^2(\alpha) \tag{5}$$

Repeating (5) recursively we get:

$$F_T \leq \left(1 - \frac{2\mu\eta}{c}(1 - \alpha^2 m)\right)^T F_0 + \left(\frac{\eta\alpha^2}{2}\zeta^2 + \frac{L\eta^2}{2}\tilde{\sigma}^2(\alpha)\right)\sum_{i=0}^{T-1}\left(1 - \frac{2\mu\eta}{c}(1 - \alpha^2 m)\right)^i$$

$$\leq \left(1 - \frac{2\mu\eta}{c}(1 - \alpha^2 m)\right)^T F_0 + \left(\frac{\eta\alpha^2}{2}\zeta^2 + \frac{L\eta^2}{2}\tilde{\sigma}^2(\alpha)\right)\frac{c}{2\mu\eta(1 - \alpha^2 m)}$$

$$= \left(1 - \frac{2\mu\eta}{c}(1 - \alpha^2 m)\right)^T F_0 + \frac{c\alpha^2}{4\mu(1 - \alpha^2 m)}\zeta^2 + \frac{cL\eta}{4\mu(1 - \alpha^2 m)}\tilde{\sigma}^2(\alpha)$$

Choosing $2(1 - \alpha^2 m)\eta/c = \min\left(\eta_{\max}, \frac{\log(\max(2, \frac{2\mu F_0 T}{3L\tilde{\sigma}(\alpha)^2}))}{2\mu T}\right)$ we get:

$$F_T = \tilde{\mathcal{O}}\left(F_0 \exp\left(-\mu\eta_{\max}T\right) + \frac{L\tilde{\sigma}(\alpha)^2}{\mu^2 T(1 - \alpha^2 m)^2} + \frac{\alpha^2}{\mu(1 - \alpha^2 m)}\zeta^2\right).$$

This concludes the proof of Theorem 1 in the $\mu$-PL case.

In the article, we argued that we can get rid of the logarithmic factors hidden in the notation $\tilde{\mathcal{O}}$. We show now how to do it for the $\mu$-PL case.

**$\mu$-PL with a decreasing step-size.** starting from (5), we choose a step size $\eta_t$ such that $1 - \frac{2\mu\eta_t}{c}(1 - \alpha^2 m) = \frac{t^2}{(t+1)^2}$, this means $\eta_t = \frac{c(2t+1)}{2\mu(1 - \alpha^2 m)(t+1)^2}$, this choice transforms (5) into

$$(t+1)^2 F_{t+1} \leq t^2 F_t + \frac{c(2t+1)\alpha^2}{4\mu(1 - \alpha^2 m)}\zeta^2 + \frac{c^2 L(2t+1)^2}{8\mu^2(1 - \alpha^2 m)^2(t+1)^2}\tilde{\sigma}^2(\alpha)$$

Summing the last inequality for $t = 0$ to $t = T-1$, and using the fact $\sum_{t=0}^{T-1} 2t+1 = T^2$ and $2t+1 \leq 2(t+1)$, we get:

$$T^2 F_T \leq \frac{cT^2\alpha^2}{4\mu(1 - \alpha^2 m)}\zeta^2 + \frac{c^2 LT}{2\mu^2(1 - \alpha^2 m)^2}\tilde{\sigma}^2(\alpha)$$

Dividing by $T^2$:

$$F_T \leq \frac{c\alpha^2}{4\mu(1 - \alpha^2 m)}\zeta^2 + \frac{c^2 L}{2\mu^2(1 - \alpha^2 m)^2 T}\tilde{\sigma}^2(\alpha)$$

This indeed is the same rate but without any hidden logarithmic factors in $T$.

To be rigorous, we need to make sure that our decreasing step-size verifies $\eta_t \leq \eta_{\max}$, this will mean we can't sum starting from $t = 0$, but instead we need to start from $t = t_0$ such that $\eta_{t_0} \leq \eta_{\max}$ is verified. Doing this will lead to

$$F_T \leq \frac{c\alpha^2}{4\mu(1 - \alpha^2 m)}\zeta^2 + \frac{c^2 L}{2\mu^2(1 - \alpha^2 m)^2 T}\tilde{\sigma}^2(\alpha) + \frac{t_0^2 F_{t_0}}{T^2}$$

In the general case where we are collaborating with N agents and using the weights $\{\tau_k\}_{k=1}^N$, as discussed before, it suffices to replace $M_1$ by $M_{\mathrm{avg}}/N$ and $\sigma_1^2$ by $\sigma_{\mathrm{avg}}^2/N$.

**Choice of the weights $\{\tau_k\}_{k=1}^N$.** Based on the $\mu$-PL bound, the best choice of the weights $\{\tau_k\}_{k=1}^N$ is given by the following constrained quadratic programming problem :

$$\min_{\tau_1 \geq 0, \ldots, \tau_N \geq 0, \sum_j \tau_j = 1} \sum_{k=1}^N \frac{L}{\mu T(1 - \alpha^2 m)}\tau_k^2\sigma_k^2 + \tau_k\zeta_k^2,$$

As $T \to \infty$, the program becomes that of minimizing the average bias i.e.

$$\min_{\tau_1 \geq 0, \ldots, \tau_N \geq 0, \sum_j \tau_j = 1} \sum_{k=1}^N \tau_k\zeta_k^2,$$

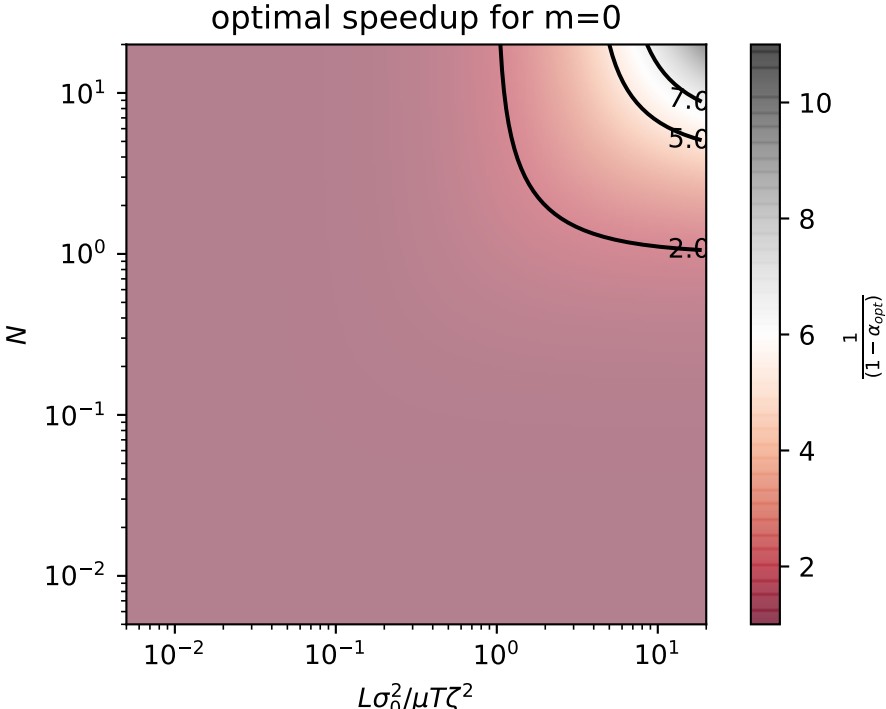

Figure 5: Collaborative training speedup factor $1/(1 - \alpha_{\mathrm{opt}})$, indicated as color, as a function of the number of collaborators $N$ (y-axis) and $\frac{L\sigma_0^2}{\mu T \zeta^2}$ (x-axis) for $m = 0$. The bigger $N$ and the smaller $T\zeta^2$ (cumulative bias) is relative to $\sigma_0^2$, the bigger the resulting speedup from collaboration.

The solution to this problem is easy, only the agents who have the smallest bias will get a non-zero weight. However, for $T$ finite, the term $\sum_{k=1}^N \tau_k^2 \sigma_k^2$ also plays a role and the weights should be taken to minimize it too. What is important is that as expected, the smaller $\zeta_k^2$ and $\sigma_k^2$ are the bigger the weight it will be given to agent $k$.

To study the effect of $N$ on the convergence rate, we will pick a middle ground where $\sigma_k^2 = \sigma_0^2$ and $\zeta_k = \zeta$ for all agents $k$.

**Choice of the collaboration weight $\alpha$.** The collaboration weight $\alpha$ is chosen as follows:

$$\alpha \in \underset{\alpha \in (0, 1/\sqrt{m})}{\arg\min} \frac{L\tilde{\sigma}(\alpha)^2}{\mu^2 T (1 - \alpha^2 m)^2} + \frac{\alpha^2}{\mu(1 - \alpha^2 m)} \zeta^2$$

For $m = 0$, which means the bias is bounded, we have $\alpha_{\mathrm{opt}} = (1 + \frac{1}{N} + \frac{\mu \zeta^2 T}{L\sigma_0^2})^{-1}$ and we obtain a speed-up $F_T = \tilde{O}\left(\frac{L\sigma_0^2}{2\mu^2 T}(1 - \alpha_{\mathrm{opt}})\right)$. The speedup factor $1/(1 - \alpha_{\mathrm{opt}})$ is illustrated in Figure 5.

We note in particular that for $\zeta^2 = 0$, the speedup is linear, and only, in this case, do we get such a speedup.

Now if $m \neq 0$ and even in the favorable case $\zeta^2 = 0$, Figure 6 shows how much we deviate from linear speedup (obtained for $m = 0$) as $m$ is different than zero.

Figure 6: Effect of $m$ (which controls the non-constant noise and is related to scaling) on the speedup of WGA when $\zeta = 0$ (avg converges to the same point as 0). The dashed line represents the linear speedup $N + 1 \mapsto N + 1$ encountered for $m = 0$ ($N + 1$ is the total number of agents including 0). We notice that as $m$ grows the speedup becomes more and more sub-linear.

### C.3 Proof of Theorem 5.1

We will use a bias oracle on only one agent. The bias oracle gives an independent noisy estimate of the true gradient bias between agent 1 and agent 0. This bias oracle is given by $\boldsymbol{c}_{t,oracle} = \nabla_{\boldsymbol{x}} f_1(\boldsymbol{x}) - \nabla_{\boldsymbol{x}} f_0(\boldsymbol{x}) + \boldsymbol{n}_{t,oracle}$ where $\boldsymbol{n}_{t,oracle}$ is an independent noise of variance $v^2$. Using such an oracle means we are working with an unbiased estimate of $\nabla_{\boldsymbol{x}} f_0(\boldsymbol{x})$ with a variance equal to $\tilde{\sigma}^2(\alpha) = (1 - \alpha)^2 \sigma_0^2 + \alpha^2 (\sigma_1^2 + v^2)$ .

Now using (3) and $L\eta \leq 1$, we get:

$$\mathbb{E}_{\xi_t}[f_0(\boldsymbol{x}_{t+1})] - f_0(\boldsymbol{x}_t) \leq -\frac{\eta}{2} \|\nabla_{\boldsymbol{x}} f_0(\boldsymbol{x}_t)\|^2 + + \frac{L\eta^2}{2}(M\|\nabla_{\boldsymbol{x}} f_0(\boldsymbol{x})\|^2 + \tilde{\sigma}^2(\alpha))$$

$$\leq \frac{\eta}{2}(-1 + LM\eta)\|\nabla_{\boldsymbol{x}} f_0(\boldsymbol{x}_t)\|^2 + \frac{L\eta^2}{2}\tilde{\sigma}^2(\alpha)$$

For $\eta \leq \frac{1}{2ML}$ we get:

$$F_{t+1} - F_t \leq -\frac{\eta}{c}\mathbb{E}\left[\|\nabla_{\boldsymbol{x}} f_0(\boldsymbol{x}_t)\|^2\right] + \frac{L\eta^2}{2}\tilde{\sigma}^2(\alpha) \tag{6}$$

Where the constant $c = 2$ if $M = 0$ and $c = 4$ otherwise.

**Non-convex case of Theorem 5.1.** We rearrange the terms in (6), sum for $t = 0$ to $t = T - 1$ and divide by $T$, we get $\forall \eta \leq \eta_{\max} := \min(\frac{1}{L}, \frac{1}{2ML})$,

$$\frac{1}{cT}\sum_{t=0}^{T-1}\mathbb{E}\left[\|\nabla_{\boldsymbol{x}} f_0(\boldsymbol{x}_t)\|^2\right] \leq \frac{F_0}{\eta} + \frac{L\eta}{2}\tilde{\sigma}^2(\alpha) .$$

Choosing $\eta = \min\left(\eta_{\max}, \sqrt{\frac{2F_0}{L\tilde{\sigma}^2 T}}\right)$, we get:

$$\frac{1}{cT}\sum_{t=0}^{T-1}\mathbb{E}\left[\|\nabla_{\boldsymbol{x}}f_0(\boldsymbol{x}_t)\|^2\right] \leq \frac{F_0}{\eta_{\max}T} + \sqrt{\frac{2LF_0\tilde{\sigma}^2(\alpha)}{T}}.$$

$\mu$-**PL case of Theorem 5.1.** We use Assumption $A2$, to have for all $\eta \leq \eta_{\max} = \min(\frac{1}{L}, \frac{1}{2ML})$,

$$F_{t+1} \leq (1 - \frac{2\eta\mu}{c})F_t + \frac{L\eta^2}{2}\tilde{\sigma}^2(\alpha). \tag{7}$$

A recurrence on (7) yields:

$$F_T \leq (1 - \frac{2\eta\mu}{c})^T F_0 + \frac{L\eta}{2}\tilde{\sigma}^2(\alpha)\sum_{i=0}^{T-1}(1 - \frac{2\eta\mu}{c})^i \leq (1 - \frac{2\eta\mu}{c})^T F_0 + \frac{cL\eta}{4\mu}\tilde{\sigma}^2(\alpha)$$

All is left is to set $2\eta/c = \min\left(\eta_{\max}, \frac{\log(\max(2, \frac{2\mu F_0 T}{3L\tilde{\sigma}(\alpha)^2}))}{2\mu T}\right)$ to get:

$$F_T = \tilde{\mathcal{O}}\left(F_0\exp\left(-\mu\eta_{\max}T\right) + \frac{L\tilde{\sigma}(\alpha)^2}{\mu^2 T}\right).$$

### C.4 Proof of Theorem 5.2

The gradient estimator used in our bias correction algorithm $\boldsymbol{g}(\boldsymbol{x}_t) := (1-\alpha_t)\boldsymbol{g}_0(\boldsymbol{x}_t) + \alpha_t\left(\boldsymbol{g}_1(\boldsymbol{x}_t) - \boldsymbol{c}_t\right)$ can be decomposed into a bias term and a noise term in the following way

$$\boldsymbol{g}(\boldsymbol{x}_t) := \nabla_{\boldsymbol{x}}f_0(\boldsymbol{x}_t) + \underbrace{\alpha\mathbb{E}[\boldsymbol{b}_t - \boldsymbol{c}_t]}_{bias} + \underbrace{\boldsymbol{n}_{t,total}}_{noise}$$

Where $\boldsymbol{b}_t = \boldsymbol{g}_1(\boldsymbol{x}_t) - \boldsymbol{g}_0(\boldsymbol{x}_t)$ is the observed stochastic gradient bias at time $t$. Using the $L$-smoothness of $f_0$ and $\eta < 1/L$, (3) would give us the following inequality:

$$F_{t+1} - F_t \leq \frac{\eta}{2}\left(-\mathbb{E}\left[\|\nabla f_0(\boldsymbol{x}_t)\|^2\right] + \alpha^2\mathbb{E}\left[\|\mathbb{E}[\boldsymbol{b}_t - \boldsymbol{c}_t]\|^2\right]\right) + \frac{L\eta^2}{2}\mathbb{E}\left[\|\boldsymbol{n}_{t,total}\|^2\right]$$

However, due to the dependence of $\boldsymbol{c}_t$ on the past, **this is not true**. For this reason, we use a different proof strategy.

We have:

$$\boldsymbol{g}(\boldsymbol{x}^t) = (1-\alpha)\boldsymbol{g}_0(\boldsymbol{x}^t) + \alpha\boldsymbol{g}_1(\boldsymbol{x}^t) - \alpha\boldsymbol{c}^t$$

Where

$$\boldsymbol{c}^t = (1-\beta)\boldsymbol{c}^{t-1} + \beta(\boldsymbol{g}_1(\boldsymbol{x}^{t-1}) - \boldsymbol{g}_0(\boldsymbol{x}^{t-1}))$$

**Descent Lemma.** Using the $L$-smoothness of $f_0$ we have :

$$f_0(\boldsymbol{x}^{t+1}) - f_0(\boldsymbol{x}^t) \leq -\eta\langle\nabla f_0(\boldsymbol{x}^t), \boldsymbol{g}(\boldsymbol{x}^t)\rangle + \frac{L\eta^2}{2}\|\boldsymbol{g}(\boldsymbol{x}^t)\|_2^2$$

Due to the dependence of $\boldsymbol{x}^t$ on $\boldsymbol{c}^t$, we cannot take the expectation inside the inner-product. However, if we condition on the past (it will be denoted $\mathbb{E}_t$) then $\boldsymbol{c}^t$ is constant and we have :

$$\mathbb{E}_t\langle\nabla f_0(\boldsymbol{x}^t), \boldsymbol{g}(\boldsymbol{x}^t)\rangle = \langle\nabla f_0(\boldsymbol{x}^t), (1-\alpha)\nabla f_0(\boldsymbol{x}^t) + \alpha\nabla f_1(\boldsymbol{x}^t) - \alpha\boldsymbol{c}^t\rangle$$

And

$$\mathbb{E}_t \|\boldsymbol{g}(\boldsymbol{x}^t)\|_2^2 = \underbrace{\sigma^2(\alpha)}_{=(1-\alpha)^2\sigma_0^2 + \alpha^2\sigma_1^2} + \|(1-\alpha)\nabla f_0(\boldsymbol{x}^t) + \alpha\nabla f_1(\boldsymbol{x}^t) - \alpha\boldsymbol{c}^t\|_2^2$$

So

$$\begin{aligned}
\mathbb{E}_t f_0(\boldsymbol{x}^{t+1}) - f_0(\boldsymbol{x}^t) &\leq -\eta\langle\nabla f_0(\boldsymbol{x}^t), (1-\alpha)\nabla f_0(\boldsymbol{x}^t) + \alpha\nabla f_1(\boldsymbol{x}^t) - \alpha\boldsymbol{c}^t\rangle \\
&\quad + \frac{L\eta^2}{2}\big(\sigma^2(\alpha) + \|(1-\alpha)\nabla f_0(\boldsymbol{x}^t) + \alpha\nabla f_1(\boldsymbol{x}^t) - \alpha\boldsymbol{c}^t\|_2^2\big) \\
&\leq \frac{\eta}{2}\big(-\|\nabla f_0(\boldsymbol{x}^t)\|_2^2 + \alpha^2\|\nabla f_1(\boldsymbol{x}^t) - \nabla f_0(\boldsymbol{x}^t) - \boldsymbol{c}^t\|_2^2\big) \\
&\quad + \frac{L\eta^2}{2}\sigma^2(\alpha)
\end{aligned}$$

Where we have used above $\eta L \leq 1$ and the identity $-2\langle\boldsymbol{a}+\boldsymbol{b}, \boldsymbol{a}\rangle + \|\boldsymbol{a}+\boldsymbol{b}\|_2^2 = \|\boldsymbol{b}\|_2^2 - \|\boldsymbol{a}\|_2^2$.

So

$$\begin{aligned}
\mathbb{E}[f_0(\boldsymbol{x}^{t+1})] - \mathbb{E}[f_0(\boldsymbol{x}^t)] &\leq \frac{\eta}{2}\big(-\mathbb{E}[\|\nabla f_0(\boldsymbol{x}^t)\|_2^2] + \alpha^2\mathbb{E}[\|\nabla f_1(\boldsymbol{x}^t) - \nabla f_0(\boldsymbol{x}^t) - \boldsymbol{c}^t\|_2^2]\big) + \frac{L\eta^2}{2}\sigma^2(\alpha) \\
&\leq \frac{-\eta}{2}\mathbb{E}[\|\nabla f_0(\boldsymbol{x}^t)\|_2^2] + \frac{L\eta^2}{2}\sigma^2(\alpha) \\
&\quad + \alpha^2\eta\mathbb{E}[\|\nabla f_1(\boldsymbol{x}^t) - \nabla f_0(\boldsymbol{x}^t) - f_1(\boldsymbol{x}^{t-1}) + \nabla f_0(\boldsymbol{x}^{t-1})\|_2^2] \\
&\quad + \alpha^2\eta\mathbb{E}[\|\boldsymbol{c}^t - f_1(\boldsymbol{x}^{t-1}) + \nabla f_0(\boldsymbol{x}^{t-1})\|_2^2]
\end{aligned}$$

Using the $\delta-$BHD assumption, we have :

$$\mathbb{E}[\|\nabla f_1(\boldsymbol{x}^t) - \nabla f_0(\boldsymbol{x}^t) - f_1(\boldsymbol{x}^{t-1}) + \nabla f_0(\boldsymbol{x}^{t-1})\|_2^2] \leq \delta^2\mathbb{E}[\|\boldsymbol{x}^t - \boldsymbol{x}^{t-1}\|_2^2] := \delta^2\Delta^t$$

We will use the notation : $E_c^t = \mathbb{E}[\|\boldsymbol{c}^t - f_1(\boldsymbol{x}^{t-1}) + \nabla f_0(\boldsymbol{x}^{t-1})\|_2^2]$, $G^t = \mathbb{E}[\|\nabla f_0(\boldsymbol{x}^t)\|_2^2]$ and $F^t = \mathbb{E}[f_0(\boldsymbol{x}^t)] - f_0^\star$.

All in all, we have :

$$F_{t+1} - F_t \leq \frac{-\eta}{2}G^t + \frac{L\eta^2}{2}\sigma^2(\alpha) + \alpha^2\delta^2\eta\Delta^t + \alpha^2\eta E_c^t \tag{8}$$

**Bounding $\Delta^t$.** We also show that :

$$\Delta^t \leq \eta^2\big(\sigma^2(\alpha) + 3G^{t-1} + 3\alpha^2\delta^2\Delta^{t-1} + 3\alpha^2 E_c^{t-1}\big) \tag{9}$$

*Proof.*

$$\begin{aligned}
\Delta^t &= \mathbb{E}[\|\boldsymbol{x}^t - \boldsymbol{x}^{t-1}\|_2^2] \\
&= \eta^2\mathbb{E}[\|\boldsymbol{g}(\boldsymbol{x}^{t-1})\|_2^2] \\
&= \eta^2\big(\sigma^2(\alpha) + \mathbb{E}[\|\nabla f_0(\boldsymbol{x}^{t-1}) + \alpha(\nabla f_1(\boldsymbol{x}^{t-1}) - \nabla f_0(\boldsymbol{x}^{t-1}) - \boldsymbol{c}^{t-1})\|_2^2]\big) \\
&\leq \eta^2\big(\sigma^2(\alpha) + 3\mathbb{E}[\|\nabla f_0(\boldsymbol{x}^{t-1})\|_2^2] \\
&\quad + 3\alpha^2\mathbb{E}[\|\nabla f_1(\boldsymbol{x}^{t-1}) - \nabla f_0(\boldsymbol{x}^{t-1}) - \nabla f_1(\boldsymbol{x}^{t-2}) + \nabla f_0(\boldsymbol{x}^{t-2}))\|_2^2] + 3\alpha^2\mathbb{E}[\|\boldsymbol{c}^{t-1} - \nabla f_1(\boldsymbol{x}^{t-1}) + \nabla f_0(\boldsymbol{x}^{t-2})\|_2^2]\big) \\
&\leq \eta^2\big(\sigma^2(\alpha) + 3G^{t-1} + 3\alpha^2\delta^2\Delta^{t-1} + 3\alpha^2 E_c^{t-1}\big)
\end{aligned}$$

$\square$

**Bounding momentum error $E_c^t$.** Using the recursive definition of $\boldsymbol{c}^t$, it is easy to prove:

$$E_c^t \leq (1-\beta)E_c^{t-1} + \frac{2\delta^2}{\beta}\Delta^{t-1} + \beta^2(\sigma_0^2 + \sigma_1^2) \tag{10}$$

*Proof.*

$$
\begin{aligned}
E_c^t &= \mathbb{E}[\|\boldsymbol{c}^t - \nabla f_1(\boldsymbol{x}^{t-1}) + \nabla f_0(\boldsymbol{x}^{t-1})\|_2] \\
&= \mathbb{E}[\|(1-\beta)\boldsymbol{c}^{t-1} + \beta(\boldsymbol{g}_1(\boldsymbol{x}^{t-1}) - \boldsymbol{g}_0(\boldsymbol{x}^{t-1})) - \nabla f_1(\boldsymbol{x}^{t-1}) + \nabla f_0(\boldsymbol{x}^{t-1})\|_2] \\
&= \beta^2(\sigma_0^2 + \sigma_1^2) + (1-\beta)^2\mathbb{E}[\|\boldsymbol{c}^{t-1} - \nabla f_1(\boldsymbol{x}^{t-1}) + \nabla f_0(\boldsymbol{x}^{t-1})\|_2] \\
&\leq \beta^2(\sigma_0^2 + \sigma_1^2) + (1-\beta)^2(1+\frac{\beta}{2})E_c^{t-1} + (1-\beta)^2(1+\frac{2}{\beta})\mathbb{E}[\|\nabla f_1(\boldsymbol{x}^{t-1}) - \nabla f_0(\boldsymbol{x}^{t-1}) - f_1(\boldsymbol{x}^{t-2}) + f_1(\boldsymbol{x}^{t-2})\|_2] \\
&\leq \beta^2(\sigma_0^2 + \sigma_1^2) + (1-\beta)^2(1+\frac{\beta}{2})E_c^{t-1} + (1-\beta)^2(1+\frac{2}{\beta})\delta^2\mathbb{E}[\|\boldsymbol{x}^{t-1} - \boldsymbol{x}^{t-2}\|_2^2] \\
&\leq (1-\beta)E_c^{t-1} + \frac{2\delta^2}{\beta}\Delta^{t-1} + \beta^2(\sigma_0^2 + \sigma_1^2)
\end{aligned}
$$

$\square$

**Non-convex case.**

Combining Inequalities 8, 9 and 10, we prove that for $\eta \leq 1/(6\alpha^2\delta^2)$ :

$$
\Phi_{t+1} - \Phi_t \leq \frac{L\sigma^2(\alpha)}{2}\eta^2 + \frac{10\alpha^2\delta^2\sigma^2(\alpha)}{\beta^2}\eta^3 + 2\alpha^2\beta\eta(\sigma_0^2 + \sigma_1^2) - \frac{\eta}{4}G^t \tag{11}
$$

For the potential $\Phi_t = F_t + \frac{2\alpha^2\eta}{\beta}E_c^t + \frac{10\alpha^2\delta^2\eta}{\beta^2}\Delta^t$.

By adding the terms in Inequality 11 from $t = 0$ to $T-1$ and by noting that $\Delta^0 \leq \eta^2(2\zeta^2 + 2(1+m)E[\|\nabla f_0(\boldsymbol{x}^0)\|^2]) := \eta^2\tilde{\zeta}^2$, we get :

$$
\frac{1}{4T}\sum_{t=0}^{T-1}G^t \leq \frac{F_0}{\eta T} + \frac{2\alpha^2}{\beta T}E_c^0 + \frac{L\sigma^2(\alpha)}{2}\eta + \frac{10\alpha^2\delta^2\eta^2}{\beta^2}(\tilde{\zeta}^2/T + \sigma^2(\alpha)) + 2\alpha^2\beta(\sigma_0^2 + \sigma_1^2)
$$

At this level, we choose $\beta \in \arg\min_{\beta \in [0,1]} \frac{10\alpha^2\delta^2\eta^2}{\beta^2}(\tilde{\zeta}^2/T + \sigma^2(\alpha)) + 2\alpha^2\beta(\sigma_0^2 + \sigma_1^2)$ this means choosing $\beta = \min(1, \left(\frac{10\delta^2(\tilde{\zeta}^2/T + \sigma^2(\alpha))}{\sigma_0^2 + \sigma_1^2}\right)^{1/3}\eta^{2/3})$. This choice gives the inequality in theorem 5.2 :

$$
\frac{1}{4T}\sum_{t=0}^{T-1}\mathbb{E}[\|\nabla f_0(\boldsymbol{x}_t)\|^2] \leq \frac{F_0}{\eta T} + \frac{4\alpha^2 E_0}{\beta T} + 12\alpha^2\left((\sigma_0^2 + \sigma_a^2)(\tilde{\zeta}^2/T + \sigma^2(\alpha))\right)^{1/3}(\delta\eta)^{2/3} + \frac{L\sigma^2(\alpha)}{2}\eta + 10\alpha^2\delta^2\sigma^2(\alpha)\eta^2 .
$$

The term $\frac{4\alpha^2 E_0}{\beta T}$ has a smaller magnitude than the term $\frac{F_0}{\eta T}$ (because $\lim_{\eta \to 0}\eta/\beta = 0$). Furthermore, using a batch $S$ times larger for estimating the first bias means that $E_0 \leq (\sigma_0^2 + \sigma_a^2)/S$.

**$\mu$-PL case.** For the $\mu$-PL case we use the fact that $2\mu(f_0(\boldsymbol{x}) - f_0^\star) \leq \|\nabla f_0(\boldsymbol{x})\|_2^2 \leq 2L(f_0(\boldsymbol{x}) - f_0^\star)$ which is equivalent (in our notation) to $2\mu F_t \leq G^t \leq 2LF_t$.

Combining this with Inequalities 8, 9 and 10 we get :

$$
\begin{aligned}
F_{t+1} &\leq (1-\eta\mu)F_t + \frac{L\eta^2}{2}\sigma^2(\alpha) + \alpha^2\delta^2\eta\Delta^t + \alpha^2\eta E_c^t \\
\Delta^t &\leq \eta^2\left(\sigma^2(\alpha) + 6LF_{t-1} + 3\alpha^2\delta^2\Delta^{t-1} + 3\alpha^2 E_c^{t-1}\right) \\
E_c^t &\leq (1-\beta)E_c^{t-1} + \frac{2\delta^2}{\beta}\Delta^{t-1} + \beta^2(\sigma_0^2 + \sigma_1^2)
\end{aligned}
$$

Combining these three inequalities we get :

$$\Phi_{t+1} \leq (1 - \frac{\mu\eta}{2})\Phi_t + \frac{L\sigma^2(\alpha)}{2}\eta^2 + \frac{10\alpha^2\delta^2\sigma^2(\alpha)}{\beta^2}\eta^3 + 2\alpha^2\beta\eta(\sigma_0^2 + \sigma_1^2) \tag{12}$$

For the same potential as in the Non-convex case.

Reiterating this inequality gives :

$$\Phi_T \leq (1 - \frac{\mu\eta}{2})^T\Phi_0 + \frac{L\sigma^2(\alpha)}{\mu}\eta + \frac{20\alpha^2\delta^2\sigma^2(\alpha)}{\mu\beta^2}\eta^2 + \frac{4\alpha^2\beta}{\mu}(\sigma_0^2 + \sigma_1^2) \tag{13}$$

At this point, we choose $\beta$ that optimizes the right-hand side in the previous inequality, and we obtain $\beta = \min(1, \left(\frac{10\delta^2\sigma^2(\alpha)}{\sigma_0^2+\sigma_1^2}\right)^{1/3}\eta^{2/3})$

We get then

$$\Phi_T \leq (1 - \frac{\mu\eta}{2})^T\Phi_0 + \frac{L\sigma^2(\alpha)}{\mu}\eta + 24\alpha^2\left((\sigma_0^2 + \sigma_a^2)\sigma^2(\alpha)\right)^{1/3}(\delta\eta)^{2/3}/\mu \tag{14}$$

To beat training alone we would need $(\delta\eta)^{2/3} \ll \eta$ which means $\delta^2 \ll \eta$. As it is known $\eta$ is of order $\frac{1}{T}$ in the $\mu$-PL case, this means we need $\delta^2 = o(\frac{1}{T})$ to beat training alone.

Choosing $\eta = \min(\eta_{max}, \frac{\log(\max(2, \frac{2\mu\Phi_0 T}{3L\sigma^2(\alpha)}))}{\mu T})$ we get :

$$\Phi_T \in \tilde{\mathcal{O}}\left(\Phi_0 \exp\left(-\mu\eta_{max}T/2\right) + \frac{L\sigma^2(\alpha)}{\mu^2 T} + 24\alpha^2\left((\sigma_0^2 + \sigma_a^2)\sigma^2(\alpha)\right)^{1/3}\delta^{2/3}/(\mu^{5/3}T^{2/3})\right)$$

**Choices of the weights.** The optimal choices of the weights $\alpha$ and $\tau_k$ are obtained by minimizing the right-hand-side of the above inequality, this will give a quadratic problem that needs to be solved under the conditions $\sum_{k=1}^{N}\tau_k = 1$ and $\tau_k \geq 0$. As $T$ goes to $\infty$, the bias $\zeta^2$ disappears and this choice is fully dictated by the variance. In fact we can simply minimize the variance $\sigma^2(\alpha) = (1 - \alpha)^2\sigma_0^2 + \alpha^2 \sum_k \tau_k^2\sigma_k^2$.

**Proof of Corollary 5.3 :**

Now supposing $\delta^2 = o(\frac{1}{\sqrt{T}})$, for example $\delta^2 = \frac{\delta_0^2}{T^{3a+1/2}}$ for some $a > 0$, then by choosing $\eta = \min(1/L, 1/(6\alpha^2\delta^2), \sqrt{\frac{2F_0}{L\sigma^2(\alpha)T}})$ we get :

$$\frac{1}{4T}\sum_{t=0}^{T-1}\mathbb{E}[\|\nabla f_0(\boldsymbol{x}_t)\|^2] \leq 3\sqrt{\frac{LF_0\sigma^2(\alpha)}{T}} + 12\alpha^2\left(\frac{\sigma_0^2 + \sigma_a^2}{\sigma^2(\alpha)}(\tilde{\zeta}^2/T + \sigma^2(\alpha))\frac{2\delta_0^2 F_0}{L}\right)^{1/3}\frac{1}{T^{1/2+a}}$$

$$+ 4\alpha^2 E_0\left(\frac{L(\sigma_0^2 + \sigma_a^2)}{10\delta^2 F_0}\right)^{1/3}\frac{1}{T^{2/3}}$$

$$+ \frac{(L + \alpha^2\delta^2 + \alpha^2\delta^2/L)F_0 + 4\alpha^2 E_0}{T}$$

We can choose $\alpha$ and the weights $\tau_k$ in such a way to optimize $\sigma^2(\alpha) = (1 - \alpha)^2\sigma_0^2 + \alpha^2 \sum_k \tau_k^2\sigma_k^2$, but we can simply choose $\alpha = \frac{N}{N+1}$ and $\tau_k = \frac{1}{N}$ this will guarantee that $\sigma^2(\alpha) = \frac{\sigma_{avg}^2}{N}$ for $\sigma_{avg}^2 = \frac{\sum_{k=0}^{N}\sigma_k^2}{N}$ is the average variance. This choice of the weights implies that the dominant order in $T$ has a linear speedup in $N$ which is the statement of Corollary 5.3.

# D   Code

The code for our experiments can be found at `https://anonymous.4open.science/r/LinSpeedUpCode-F695`.

