# OpenReview forum: "Linear Speedup in Personalized Collaborative Learning"
_TMLR — Rejected by TMLR_

### Review · Reviewer_WBzE · 2022-12-02

**Summary Of Contributions:**

This paper studies collaborative learning by stochastic optimization. In this framework, we want to solve a task $0$ given access to $N$ related tasks $1,2,\ldots,N$. At each iteration, each task builds a stochastic gradient based on its own dataset and sends this stochastic gradient to task $0$. The task $0$ then performs an aggregation to build a stochastic gradient used to update the weight. The paper considers two gradient aggregation schemes. The first is the weighted gradient averaging (WGA), which collaborates weights are given to different local gradients. The second is the bias correction scheme to remove the bias in WGA. Convergence rates are developed for both WGA and EMA for nonconvex problems with or without PL conditions. Experimental results are reported to compare the behavior of different algorithms and to verify the theory.

**Audience:**

Yes

**Broader Impact Concerns:**

Broader Impact Concerns are not applicable.

**Claims And Evidence:**

Yes

**Requested Changes:**

I would request the authors to check the correctness of the theoretical analysis.

I would also request the authors to give some explanations on the challenge to relax the conditions for linear speedup.

I would suggest the authors to consider more challenging problems in the experiments.

Some typos also need to be addressed in the revision:

* return $x_T$ in Alg 1 should be put in a separate line

* Section 5: collaborateWe

* Section 4: $E[F_0]$ should be $\mathbb{E}[F_0]$

* Eq (3): $++$ should be $+$

**Strengths And Weaknesses:**

**Strength**

The theoretical analysis shows that the proposed algorithm attains a linear speed up, i.e., the number of required iterations decreases by a factor of N. The idea of bias correction is interesting. The theoretical analysis shows the effect of bias and variance.

**Weakness**

The linear speed up can only be achieved in very restricted cases. For WGA, the lienar speed up can only be obtained for $m=0$ and $\xi=0$. This is limited since it requires $f_k=f_0$. For EMA, the linear speed up can only be achieved if $\delta^2=o(1/\sqrt{T})$. Note $T$ is the number of iterations, then this requires that the Hessian matrices for different tasks are very similar.

The theoretical analysis seems to be standard. Furthermore, there are some issues in the analysis. In the proof of Eq (3), the expectation operators should be conditional expectation operators. Otherwise, the expectation operators cannot be moved inside the inner product. The proof of Thm 4.1 seems to consider only the case $N=1$. For example, the notation $n(x,\xi_t)=(1-\alpha)n(x,\xi_t^{(0)})+\alpha n_1(x,\xi_t^{(1)})$ and the notation $b(x_t)=\alpha (\nabla_xf_1(x_t)-\nabla_xf_0(x_t))$ only correspond to the case $N=1$. I would suggest the authors check the details.

The experimental analysis considers $1$D problems, which seem to be too simple. It would be interesting if the authors can consider more challenging problems in the experiments.

---

### Review · Reviewer_Bvok · 2022-12-11

**Summary Of Contributions:**

This paper studies personalized ML where a particular user's model training is helped by N other users. The authors first studied weighted gradient averaging (WGA) and presented its convergence result under certain assumptions. An important bias-variance tradeoff is then revealed, which motivated the bias correction variant. The analysis of the bias correction algorithm starts with the ideal case of a bias oracle, and then uses an exponential moving average to approximate the oracle. Experiments on a single noisy quadratic optimization problem corroborated the analysis.

**Audience:**

Yes

**Broader Impact Concerns:**

Nothing stands out as concerns.

**Claims And Evidence:**

Yes

**Requested Changes:**

See the weaknesses above. More experiment results using real-world dataset would be a required change from me.


**Strengths And Weaknesses:**

Strengths
+ The proposed algorithms are interesting.
+ The theoretical results are strong, with good intuition and insight.
+ The paper is well written.

Weaknesses
- The gradient similarity assumption is critical to the algorithm and the theory. However, this assumption is very strong in the sense that it has to hold for all possible models x, at all clients. This is a significant constraint on the statistical heterogeneity among users, which is very different than typical federated learning. In reality, how does one know whether this assumption holds or not? Who is going to make the decision whether a particular user n can help user 0?
- The entire work focuses on user 1, 2, ... N helping user 0. It is not a symmetric case: what about every user wants to help other users, while also being helped by other users simultaneously?
- The setting as it stands is similar to the following FL case: a central server (user 0) wants to train a local model x_0 by leveraging the data at N clients (user 1 to N), and the server also has some local data to train its own model. This setting has been studied in [R1] below but entirely in an IID setting. The current paper can be viewed as more of a non-IID setting. It would be interesting to compare and discuss the connections and differences.
- I understand that as a theory-focused paper, the experimental aspect may not be the focus of this work. However, I'd like to see at least some real-world dataset experiment results reported in the paper. I believe this is very important to get some sense whether the proposed methods can empirically have strong performance even if the assumptions do not hold.

[R1] K. Yang and C. Shen, "On the Convergence of Hybrid Federated Learning with Server-Clients Collaborative Training," 2022 56th Annual Conference on Information Sciences and Systems (CISS), 2022, pp. 252-257, doi: 10.1109/CISS53076.2022.9751161.

---

### Review · Reviewer_nSJu · 2023-02-04

**Summary Of Contributions:**

This paper provides theoretical analysis to collaborative learning under two gradient aggregation policies: weight averaging and bias correction, where the latter adopts a control variate. Based on the derived theory, the authors also give insights on how bias correction can enjoy linear speed up over the number of agents. Empirically, simulation results on a single quadratic problem are given.

**Audience:**

Yes

**Broader Impact Concerns:**

No foreseeable ethical issue.

**Claims And Evidence:**

Yes

**Requested Changes:**

* Please elaborate why using gradients from different functions $f_k$ can help solve Equation (1).
* Please elaborate on how bias correction is fundamentally different from algorithms like SCSG (https://arxiv.org/abs/1706.09156).
* Please consider adding at least one real-world application in the experiment.

**Strengths And Weaknesses:**

Strengths
* The paper provides theoretical analysis to collaborative learning over two gradient aggregation techniques. The motivation for these techniques make sense and are important to distributed learning in general.
* The paper gives detailed discussion on how heterogeneity terms would adapt under different bounds.

Weakness
* The authors propose using various objectives rather than a single one to solve Equation (1). However, I'm not sure if, in that case, Equation (1) is still the correct objective. Indeed, if assumptions are made on the similarities of $f_k$ and $f_0$, it's not surprising the noise can be bounded. But it is not clear to me how averaging the gradients from different objectives would help solve Equation (1).
* It is not clear how the bias correction technique is fundamentally different from SCAFFOLD or SARAH (https://arxiv.org/abs/1703.00102). The authors provide explanation in A.2, saying that the main idea of SVRG-like algorithm gives unbiased gradient estimate of $g(x)$. This is somehow partially true as there are other variants like SCSG (https://arxiv.org/abs/1706.09156) already proved convergence with noisy, biased control variate. The author then elaborate the bias comes from collaborating with other agents, this goes back to the problem of previous weakness.
* The experiments are too small scale and only synthetic. I understand this is a theory paper, but validation on at least one real-world problem would definitely make this stronger.
* Some of the figures (e.g. Figure 2) are not properly placed so that they overlap with some of the text.

---

### Review · Reviewer_mieu · 2023-02-11

**Summary Of Contributions:**

The authors examine the conditions under which a given agent can benefit from personalized collaborative learning. They consider an idealized scenario in which the goal is to optimize a given user's decision rule while gaining access to the stochastic gradients of $N$ agents (communication bandwidth constraints and privacy issues are not considered). The key question is how much an agent can benefit from such "extra" information. The authors first explore a simple weighted gradient averaging strategy using a weighted average of gradient estimates as a pseudo-gradient. The authors show that while there are scenarios in which this simple strategy is sufficient, it can also lead to significant bias. The authors then present a bias correction method that uses past observed gradients to estimate and correct for these biases. It is shown that the resulting solution solves the problems of WGA with bias.
In addition, a linear speedup in the number of agents satisfying a mild dissimilarity constraint is found.

**Audience:**

Yes

**Broader Impact Concerns:**

No ethical implication

**Claims And Evidence:**

Yes

**Requested Changes:**

-   Compare and discuss the connections and differences with Yang, Kun, Shengbo Chen, and Cong Shen. "On the Convergence of Hybrid Server-Clients Collaborative Training." IEEE Journal on Selected Areas in Communications (2022).
- Explain how the bias correction differs from that used in the federated case [e.g. Scaffold]
- As the mathematical results are very incremental [basically, we redo the federated case by assigning local data parameters to the server, which is not very original], I think it is essential to motivate this paper with "concrete" applications.


**Strengths And Weaknesses:**

(+)
The paper is well-written, and the derivations are easy to follow.
(-)
- the $\delta$ -Bounded Hessian Dissimilarity,  and the gradient similarity assumption is very strong and seems to be mostly ad-hoc. This assumption is stronger than the one used to model statistical heterogeneity in deep learning.  The explanations given to justify this hypothesis are very far-fetched... It's hard to believe that the proposed conditions are of any interest at all!
-  The setting is similar to hybrid FL in which a central server (user 0) train a local model x_0 using data from N clients (users 1 to N), and local data. This situation has already been studied in  Yang, Kun, Shengbo Chen, and Cong Shen. "On the Convergence of Hybrid Server-Clients Collaborative Training." IEEE Journal on Selected Areas in Communications (2022). Basically, this is the classic federated case in which local data is added to the parameter server. One can imagine that this does not significantly modify the derivations
- For WGA, the linear speed up can be achieved only for $m=0$ and $\xi=0$. This is rather restrictive ! since $f_k=f_0$ is required. For EMA, the linear speed up is achieved when $\delta^2=o(1 / \sqrt{T})$ where $T$ is the number of iterations. This is also very restrictive

---

### Decision · Action_Editors · 2023-02-20

**Recommendation:** Reject

**Comment:**

The reviewers pointed out that this paper has the following issues:
1. The assumption on the bounded dissimilarity seems to be too strong and ad-hoc (i.e., used to facilitate theoretical analysis).
2. A comparison with several important related works is missing (see the reviews for details).  Without such a discussion, it is difficult to measure the contribution of this paper.
3. It is not clear how the bias correction technique is fundamentally different from SCAFFOLD or SARAH
4. There are not any real-world applications or datasets in the experimental study.
5. Linear speed up can only be achieved on restrictive parameters.

Unfortunately, the authors did not give any response to reviewers' questions and comments.

**Audience:**

People who are interested in Federated Learning may find this paper relevant.  However, due to (1) the strong and somewhat ad-hoc assumptions on bounded dissimilarity and (2) inadequate experimental study, it is unclear that the proposed algorithm would be useful in practice.

**Claims And Evidence:**

The theoretical results of this paper are supported by proofs.  One reviewer pointed out that the proof of Theorem 4.1 seems to consider only the case $N = 1$.  Unfortunately, the authors did not give any response to this question.

The experimental study of this paper is not sufficient.  Without any real-world datasets and applications, it is hard to convince the readers that the proposed algorithms are useful in practice.